# SARS-CoV-2 Nsp1 cooperates with initiation factors EIF1 and 1A to selectively enhance translation of viral RNA

Ranen Aviner[1,2,3]�éc, Peter V. Lidsky[1]é, Yinghong Xiao[1], Michel Tassetto[1], Damian Kim[2], Lichao Zhang[4], Patrick L. McAlpine[4], Joshua Elias[4], Judith Frydman[3]*, Raul Andino[1]*

**1** Department of Microbiology and Immunology, University of California, San Francisco, San Francisco, California, United States of America, **2** Chan Zuckerberg Biohub–San Francisco, San Francisco, California, United States of America, **3** Department of Biology and Department of Genetics, Stanford University, Stanford, California, United States of America, **4** Chan Zuckerberg Biohub–San Francisco, Stanford, California, United States of America

☉ These authors contributed equally to this work.
* ranen.aviner@czbiohub.org (RAv); jfrydman@stanford.edu (JF); raul.andino@ucsf.edu (RAn)

**Data Availability Statement:** Sequencing data were deposited in SRA database under BioProject number PRJNA932822. The mass spectrometry proteomics data have been deposited to the

## Abstract

A better mechanistic understanding of virus-host dependencies can help reveal vulnerabilities and identify opportunities for therapeutic intervention. Of particular interest are essential interactions that enable production of viral proteins, as those could target an early step in the virus lifecycle. Here, we use subcellular proteomics, ribosome profiling analyses and reporter assays to detect changes in protein synthesis dynamics during SARS-CoV-2 (CoV2) infection. We identify specific translation factors and molecular chaperones that are used by CoV2 to promote the synthesis and maturation of its own proteins. These can be targeted to inhibit infection, without major toxicity to the host. We also find that CoV2 non-structural protein 1 (Nsp1) cooperates with initiation factors EIF1 and 1A to selectively enhance translation of viral RNA. When EIF1/1A are depleted, more ribosomes initiate translation from a conserved upstream CUG start codon found in all genomic and subgenomic viral RNAs. This results in higher translation of an upstream open reading frame (uORF1) and lower translation of the main ORF, altering the stoichiometry of viral proteins and attenuating infection. Replacing the upstream CUG with AUG strongly inhibits translation of the main ORF independently of Nsp1, EIF1, or EIF1A. Taken together, our work describes multiple dependencies of CoV2 on host biosynthetic networks and proposes a model for dosage control of viral proteins through Nsp1-mediated control of translation start site selection.

## Author summary

Host-directed antivirals offer a promising therapeutic approach for many viruses, including SARS-CoV-2 (CoV2), but their development requires a deeper understanding of virus-host interactions. Of particular interest are interactions that selectively promote the synthesis of viral proteins, including RNA-binding proteins, translation factors and

ProteomeXchange Consortium via the PRIDE partner repository with the dataset identifier PXD039981. https://www.ncbi.nlm.nih.gov/bioproject/PRJNA932822 https://www.ebi.ac.uk/pride/archive/projects/PXD039981/.

**Funding:** This work was supported by NIH grants AI171421, AI137471, AI169460 to R.An. and NIH grants AI127447 to JF. The funders did not play any role in the study design, data collection and analysis, or the decision to publish the study and preparation of the manuscript.

**Competing interests:** The authors have declared that no competing interests exist.

molecular chaperones. Drugs that inhibit such biosynthetic factors have already entered clinical trials for multiple indications. To identify new cellular targets for intervention, we isolated translating ribosomes from CoV2-infected and control cells and analyzed their interacting partners by mass-spectrometry. We found multiple biosynthetic factors that were specifically enriched on polysomes translating CoV2, including translation initiation factors EIF1 and 1A. These factors control translation start site selection, cooperate with the viral non-structural protein 1 (Nsp1) to selectively enhance translation of viral genomic RNA, and are exploited by CoV2 to regulate the timing and stoichiometry of viral protein synthesis. Targeting EIF1A by siRNA reduces infection with minimal toxicity to the host. Although the nature of interactions between EIF1/1A and Nsp1 remains unclear, this interdependency may provide a new strategy for antiviral therapy.

## Introduction

The process of viral replication relies heavily on the host translation machinery to produce viral proteins, replication complexes and, ultimately, infectious progeny. Interactions between viral RNA and ribosomes are therefore critical for successful infection, and most viruses have evolved mechanisms to actively hijack and usurp their host protein homeostasis (proteostasis) networks [1]. This serves two important functions: to increase the production of new virions and decrease the production of host antiviral factors. Small molecule inhibitors of e.g. translation factors and molecular chaperones have shown antiviral effects across a wide range of viruses and animal models [2], and some have entered clinical trials, suggesting that modulation of proteostasis could be a promising therapeutic approach for viral infections. For CoV2, preclinical studies have reported antiviral efficacy for drugs targeting translation initiation factor 4A1 (EIF4A1) and elongation factor 1A (EEF1A) [3,4].

We previously demonstrated that proteomic analysis of ribosome-interacting proteins in cells infected with polio, Zika or dengue viruses can yield mechanistic insights into potential host targets for antiviral intervention [5]. Here, we use a similar approach to characterize CoV2 interactions with host ribosomes. We separate translating and non-translating ribosomes from infected and control cells, and analyze their interacting proteins by liquid chromatography tandem mass-spectrometry (LC-MS/MS). We show that changes in polysome interactors reflect specific biosynthetic requirements of CoV2. RNA- and nascent chain-binding proteins, including RNA helicases, translation factors and molecular chaperones, are recruited to polysomes to support viral protein production, and their inhibition attenuates infection. Furthermore, we find that Nsp1 cooperates with initiation factors EIF1 and 1A to bypass a repressive CUG start codon in the 5' untranslated region (UTR) of CoV2 and stimulate the synthesis of CoV2 proteins. Thus, Nsp1 acts as a molecular switch that controls the relative translation of viral ORFs and modulates the stoichiometry of viral proteins.

## Results

To identify host factors involved in CoV2 protein production, we infected Vero E6 cells with SARS-CoV-2 USA/WA1/2020 at a multiplicity of infection (MOI) of 5 plaque forming units (pfu) per cell, and performed subcellular fractionation coupled to mass-spectrometry analysis (Fig 1A). Infected and control cells were lysed, fixed with formaldehyde, ultracentrifuged on 10–50% sucrose gradients, and fractionated with continuous monitoring of ribosomal RNA (rRNA) absorbance. As previously reported [6–9], CoV2 infection reduced global translation, and a lower translation rate was steadily maintained from 6 to 24 hours post-infection (hpi,

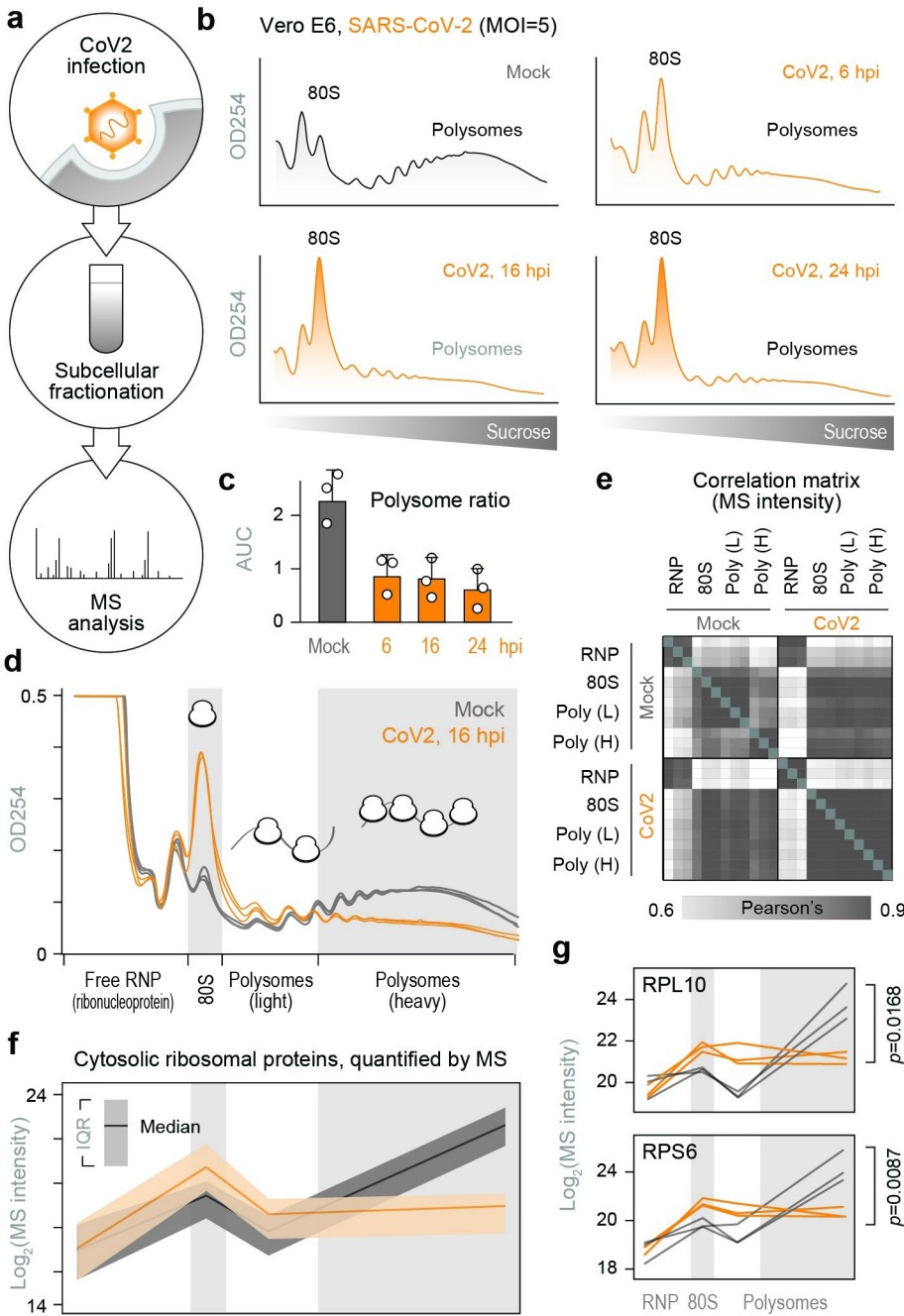

**Fig 1. Subcellular proteomics of CoV2 infected cells.** (**a**) To identify host biosynthetic networks involved in CoV2 infection, we infected Vero cells with SARS-CoV-2 USA/WA1/2020 at MOI = 5, lysed them, fixed the clarified lysates with formaldehyde and fractionated on 10–50% sucrose gradients. Crosslinking was reversed and protein content was analyzed by liquid chromatography tandem mass-spectrometry (LC-MS/MS). (**b-c**) Global protein synthesis is persistently attenuated during infection. Cells infected as above were lysed, fixed with formaldehyde and fractionated on 10–50% sucrose gradients with continuous monitoring of rRNA absorbance (b). Ratio of polysomes to sub-polysomes, calculated as the area under the curve (AUC) of relevant fractions. Shown are means±SD of 3 independent replicates (c). (**d**) Cells were infected and fractionated as above, in triplicates, and protein content was extracted from fractions containing free small (40S) and large (60S) ribosomal subunits (free RNP); 80S monosomes; and two polysome fractions ("Light" and "Heavy"). Each line reflects a single replicate, and fractions pooled for MS are indicated at bottom. (e) Correlation matrix of all host proteins identified by MS in each of the pooled fractions from either CoV2-infected or control cells. (**f**) Median and interquartile range (IQR) of all cytosolic ribosomal proteins quantified by MS in each pooled fraction from all 3 replicates. (**g**) Line plots of individual ribosomal proteins quantified by MS in each pooled fraction. Each line represents a single replicate. P, two-tailed Student's t-test p-value of differences in indicated protein abundance in heavy polysome fractions.

Fig 1B and 1C). A similar reduction was also measured in Calu3 cells (S1A Fig). Given that synthesis of viral proteins in Vero cells peaks between 10 and 24 hpi (S1B Fig) [10], we chose 16 hpi for proteomic analysis. Lysates were subjected to sucrose gradient ultracentrifugation as above, and rRNA absorbance was used to determine how to pool fractions containing free 40S and 60S ribosomal subunits (free ribonucleoprotein complexes, "RNP"), 80S monosomes, and two polysome fractions corresponding to < and ≥ 4 ribosomes per polysome ("light" and "heavy" polysomes, respectively) (Fig 1D). Based on measurements of protein concentration, the RNP fractions contained >90% of the protein mass present in the input lysates (S1C Fig). The protein content of each pooled fraction was then analyzed by label free LC-MS/MS (S1 Table). Overall, the composition of host proteins detected by MS was more variable between different fractions of the same gradient than between identical fractions of gradients from infected and uninfected cells (Fig 1E). For example, higher correlations were measured between RNP fractions of infected and uninfected cells than RNP and any other fraction across the entire dataset (Fig 1E). Given that gradient fractions separate cytoplasmic content based on size, this likely reflects common differences in composition between the monomeric proteome and larger protein assemblies. The abundance of ribosomal proteins in each pooled fraction, as quantified by MS (Figs 1F, 1G and S1D), was consistent with the observed rRNA absorbance profiles (Fig 1D). Although our lysis conditions should disrupt the membrane-enclosed coronavirus replication-transcription complex (RTC) [11], CoV2 RNA-dependent RNA-polymerase (RdRp, Nsp12) was still detected in the heavier fractions of the gradients (S1E Fig), possibly due to stabilization of large molecular weight complexes by formaldehyde crosslinking. Nevertheless, MS estimates suggest that ribosomes outnumber RdRps in these fractions by a factor of about 200 to 1.

We next performed pairwise comparisons of non-ribosomal host proteins detected in each fraction of infected and control cells. CoV2 infection had little impact on the composition of free RNP and 80S fractions, but significantly altered the protein content of heavy polysome fractions (Fig 2A). Many of the proteins recruited to heavy polysome fractions during CoV2 infection were previously shown to interact with either genomic or subgenomic CoV2 RNA [12] (gRNA/sgRNA; Fig 2A, orange). These are enriched for annotations involving RNA metabolism and protein synthesis, including RNA splicing and transport, translation, protein folding, proteasomal degradation and antigen presentation (Fig 2B and S2 Table). As previously reported, these co-translational interactions are likely mediated by both viral RNA and nascent polypeptide chains [5]. Individual examples include splicing factors RTCB, NONO and PPP1R8—three of the top 15 hits in a CRISPR screen for essential CoV2 host factors [12] —as well as SRSF5 and HNRNPD, previously shown to affect RNA splicing, translation and stability in other viruses [13,14] (Figs 2C and S2). Additional host factors recruited to CoV2 polysomes include all subunits of the 26S proteasome (Fig 2C), potentially reflecting increased co-translational degradation of nascent polypeptide chains that are sensed as aberrant by host protein quality controls [15]. Other components of the ubiquitin-proteasome system were also recruited to polysomes during infection, including HSPA5/BiP (the ER-resident Hsp70) and DNAJC3/Erdj6, which facilitate co-translational degradation of polypeptides on the ER (reviewed in [16]); and TRIM25, an E3 ligase of the antiviral ISG15 conjugation system, which targets nascent viral proteins [17] (S2 Fig).

To determine whether knowledge of polysome remodeling during CoV2 infection can provide information on druggable virus vulnerabilities, we searched for molecular chaperones and other components of proteostasis, which are critical for successful production of functional proteins across multiple virus families [2]. We infected Vero cells with CoV2 at MOI = 0.5 in the presence of either DMSO or drugs that target select host factors recruited to heavy polysome fractions during infection (Fig 3A). These include Juglone, an inhibitor of

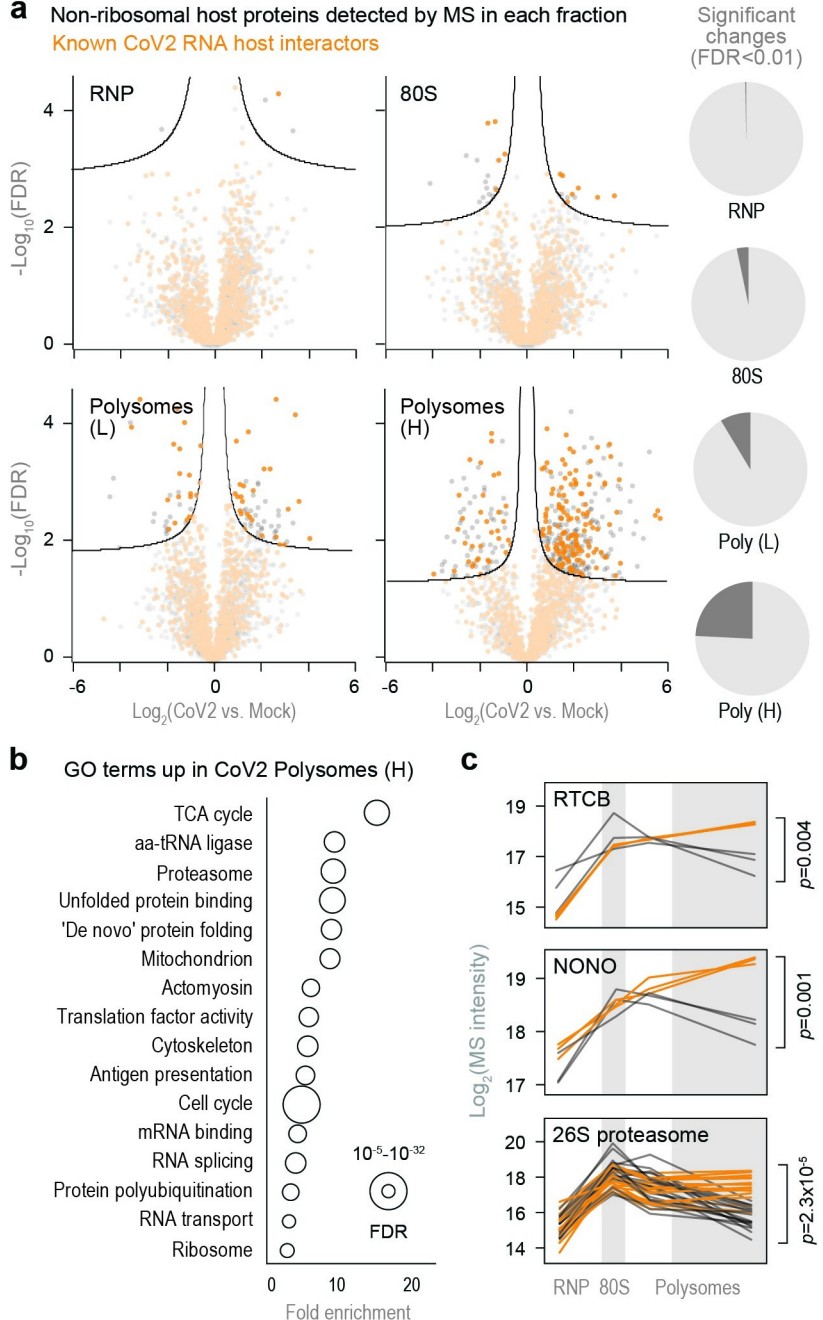

**Fig 2. CoV2 infection remodels host biosynthetic complexes.** (**a**) Pairwise comparisons of differences in individual protein abundance upon CoV2 infection of Vero cells, in each pooled fraction, as quantified by MS. Right, proportion of proteins showing statistically significant differences (FDR<0.05, S0 = 0.1) between infected and control cells. (**b**) Gene Ontology terms enriched in heavy polysome fractions from infected versus control cells. (**c**) Line plots of individual proteins quantified by MS in each fraction. Each line represents a single replicate. P, two-tailed Student's t-test p-value.

prolyl isomerase of the parvulin family; 16F16, a protein disulfide isomerase inhibitor; JG40 and JG345, which inhibit Hsp70 chaperones; and Nimbolide, which inhibits RNF114 E3 ligase. Remdesivir, a known inhibitor of the viral RdRp, was used as a positive antiviral control. At concentrations not toxic to host cells, all drugs inhibited CoV2 infection (Figs 3B, S3A and S3B),

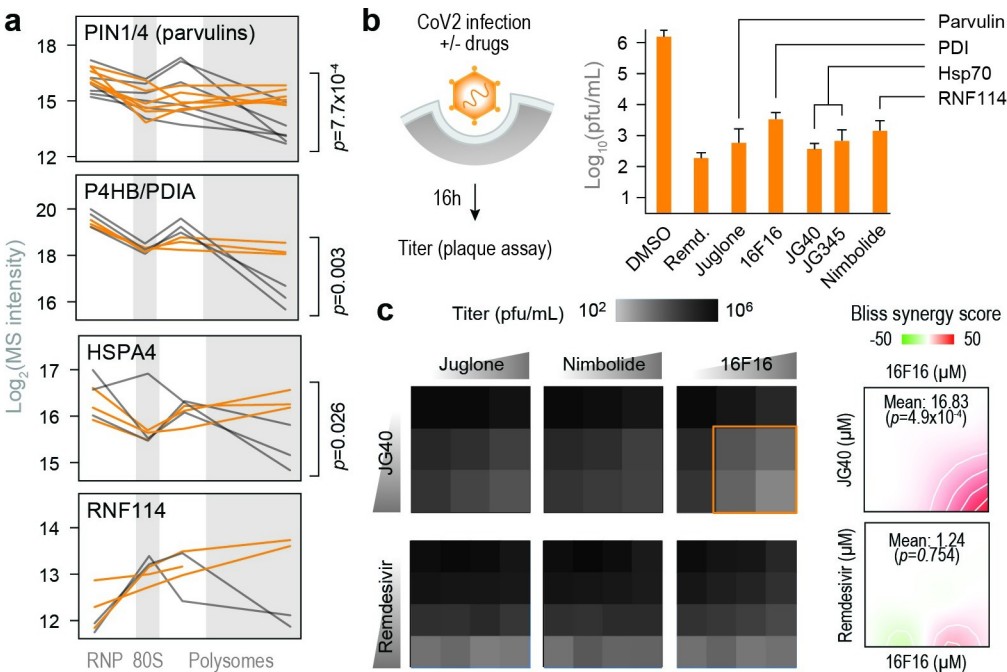

**Fig 3. Remodeling of biosynthetic complexes reveals druggable host targets for antiviral therapies.** (**a**) Line plots of individual proteostasis factors quantified by MS in each fraction. Each line represents a single replicate. P, two-tailed Student's t-test p-value. (**b**) Cells were infected with CoV2 at MOI = 0.5. Drugs were added at the start of infection, and titers were determined by plaque assays at 16 hours post-infection. Shown are means±SD of 3 independent replicates. Remdesivir, 5 μM; Juglone, 4 μM; 16F16, 2 μM; JG40, 5 μM; JG345, 5 μM; Nimbolide, 1 μM. (**c**) Cells were infected as above and treated with the indicated drug combinations. JG40, 0, 2.5, 5 μM; Remdesivir, 0, 2.5, 5, 10 μM; Juglone, 0, 0.25, 0.5, 1 μM; Nimbolide, 0, 0.25, 0.5, 1 μM; 16F16 0, 1, 2, 4 μM. Heatmaps (left) show means of 3 independent replicates. Bliss synergy plots (right) report on combination treatment with 16F16 and either JG40 or remdesivir. Higher values indicate synergistic effects.

suggesting that the virus is hyper-dependent on these targeted functions. Furthermore, combined inhibition of disulfide isomerases and Hsp70, but not disulfide isomerases and remdesivir or any other combination, had synergistic antiviral effects (Fig 3C). Together, these observations confirm that the information generated by proteomic analysis of polysomes can be used to develop antiviral interventions targeting key steps in the viral life cycle.

## Translation initiation on CoV2 gRNA is non-optimal

Next, we examined the core components of protein synthesis. Consistent with a global translation shutoff, heavy polysome fractions from infected cells contained fewer elongating ribosomes, reflected by lower abundance of ribosomal proteins and the two major elongation factors, EEF1A and EEF2 (Fig 4A). In contrast, a subset of translation initiation factors was enriched in heavy polysome fractions (Fig 4A), but not other fractions (S4A Fig), upon CoV2 infection. These included EIF1, 1A, 3A, 4A1, 4B and RNA helicase DDX3 (Figs 4A, 4B and S4B). By analyzing previous RNA interactome studies, we found that translation initiation factors are some of the most abundant host interactors of CoV2 RNA in both Vero [12] (Fig 4C) and Huh7 [18] (S4C Fig). Viral RNAs were also associated with more 40S than 60S ribosomal subunits (Figs 4C and S4C), likely representing pre-initiation complexes that await large subunit joining. If assembly of elongation-competent 80S ribosomes is slower in CoV2-infected cells, more pre-initiation complexes should accumulate in the polysome fractions. Indeed, we find more 40S than 60S ribosomal proteins in the heavy polysome fraction of infected Vero

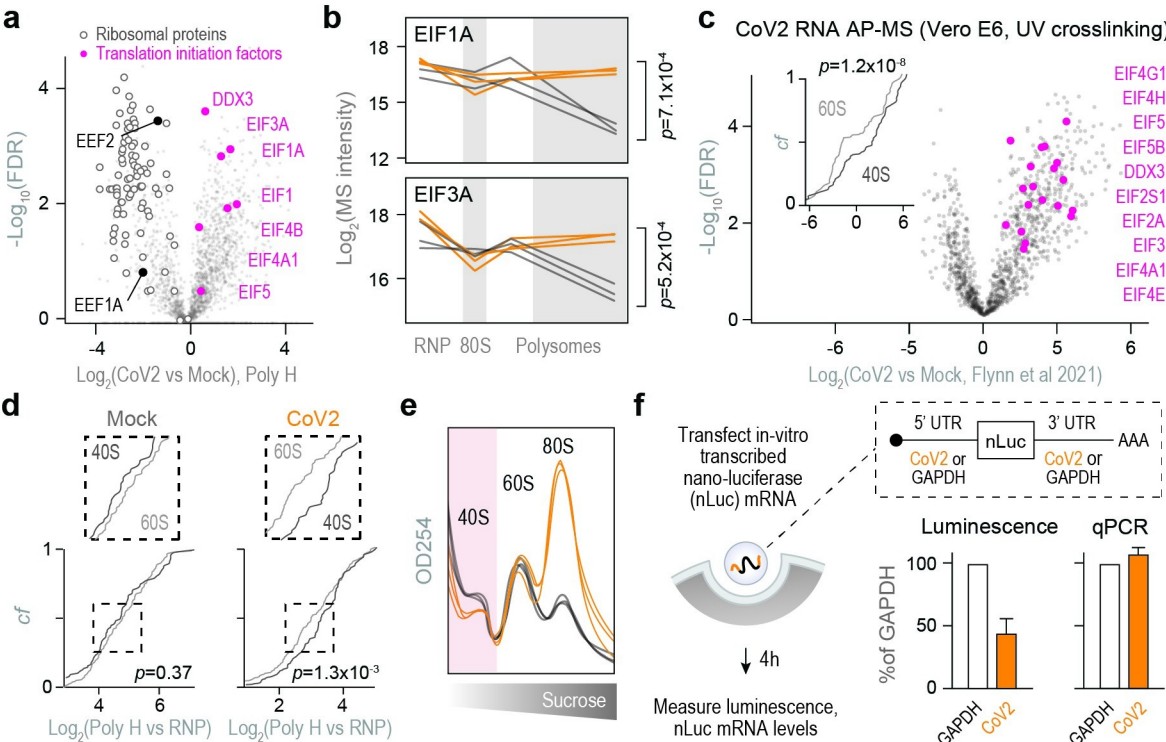

**Fig 4. Translation initiation on CoV2 gRNA is non-optimal.** (**a**) Pairwise comparisons of individual protein abundance in the heavy polysome fractions. Ribosomal proteins and translation elongation factors are depleted, whereas translation initiation factors are enriched. (**b**) Line plots of individual translation initiation factors quantified by MS in each fraction. Each line represents a single replicate. P, two-tailed Student's t-test p-value. (**c**) Translation initiation factors are highly represented in the CoV2 RNA interactome. Shown are pairwise comparisons of individual host protein abundance, quantified by MS, that specifically interact with either genomic or subgenomic CoV2 RNA during infection. Inset, cumulative distribution plots of 40S and 60S ribosomal protein interaction with CoV2 RNA. *P*, Mann-Whitney p-value. (**d**) Heavy polysome fractions contain more 40S ribosomal proteins in infected cells. Shown are cumulative distribution plots of 40S and 60S ribosomal proteins in heavy polysome fractions from infected and uninfected cells, across three replicates. *P*, Mann-Whitney p-value. (**e**) rRNA absorbance profiles from Fig 1D, showing lower abundance of free 40S subunits during CoV2 infection. (**f**) Translation initiation from CoV2 5' untranslated region (UTR) is less efficient than GAPDH 5'UTR. mRNA encoding for nano-luciferase (nLuc) flanked by 5' and 3'UTRs of either CoV2 (orange) or GAPDH (white) was transcribed in vitro, capped/polyadenylated, and transfected into Vero cells. At 4 hours post-transfection, luminescence was measured in parallel with qPCR using oligonucleotides specific to nLuc. Shown are means±SD of 3 independent replicates.

cells (Fig 4D). Furthermore, rRNA absorbance profiles of infected cells show lower abundance of free 40S but not 60S subunits in both Vero (Fig 4E) and Calu3 (Fig S4D). These observations suggest that the conversion of scanning pre-initiation complexes to initiating 80S ribosomes is inefficient—and perhaps an important rate limiting step—during CoV2 infection.

To determine whether this reflects a lower rate of translation initiation on CoV2 RNA, we generated in-vitro transcribed, capped and polyadenylated nano-luciferase (nLuc) mRNAs flanked by 5' and 3' UTRs of either GAPDH or CoV2 gRNA [9]. GAPDH UTRs were chosen as reference based on their prior use as controls for efficient constitutive translation, including in the context of CoV2 infection [9,19,20]. As the two mRNA variants encode for an identical protein sequence, differences in luminescence should reflect the relative translation initiation rate from each set of UTRs. At 4h post-transfection in Vero cells, we measured nLuc mRNA levels by RT-qPCR and protein levels by luminescence. Despite similar levels of intracellular mRNA for both reporters, CoV2 UTRs generated about 2-fold less luminescence than GAPDH UTRs (Fig 4F). Thus, translation initiation on CoV2 gRNA is less efficient than GAPDH in uninfected Vero cells. These observations are consistent with previous measurements of translation

efficiency by ribosome profiling, which showed that translation of CoV2 RNAs, and particularly gRNA, is less efficient than that of most cellular mRNAs [21,22].

## Nsp1 promotes translation initiation on CoV2 gRNA

Next, we plotted the sedimentation patterns of individual CoV2 proteins in our gradients. Non-structural proteins 2–13 shared a similar pattern of sedimentation in 80S monosome and polysome fractions (Figs 5A and S5A), likely reflecting their presence in formaldehyde-stabilized protein complexes. Such interactions between viral proteins were reported in affinity purification (AP) MS studies of CoV2 and related coronaviruses [23,24]. Nsp1, on the other hand, peaked in the subpolysome fractions (Fig 5A). Nsp1 is known to inhibit translation initiation of cellular mRNAs by blocking the ribosomal mRNA channel on 40S subunits and 80S ribosomes [6,8,25], and its accumulation in fractions containing 40S and 80S ribosomes is therefore consistent with the reported inhibitory function.

Still, Nsp1 was not entirely excluded from polysome fractions of infected cells (Fig 5A). Previous studies have shown that translation of CoV2 RNAs not only escapes repression by Nsp1 but is even enhanced by it, pointing towards additional modes of interaction with the translation machinery. Reporter assays found that low levels of Nsp1 stimulated translation from CoV2 5'UTR in transfected cells [26], and ribosome profiling analyses reported that loss of Nsp1 activity in the context of live virus infection reduced translation efficiency of viral RNAs [22]. Therefore, we next compared the effect of Nsp1 and other CoV2 proteins on translation of our nLuc reporter mRNAs. We transfected Vero cells with plasmids encoding CoV2 proteins, and at 48 h transfected again with nLuc mRNAs flanked by either GAPDH or CoV2 UTRs (Fig 5B). Most viral proteins had negligible effects on nLuc translation from either GAPDH or CoV2 UTRs (Fig 5C). In contrast, Nsp1 decreased nLuc translation from the GAPDH UTRs but increased translation from the CoV2 UTRs (Fig 5C). Given the potential destabilizing effects of Nsp1 on non-viral mRNAs [6,22], we measured the total intracellular levels of GAPDH, ACTB, and nLuc mRNA in the presence of either GFP or Nsp1. Expression of Nsp1 reduced the levels of endogenous GAPDH and ACTB (S5B Fig) but had no effect on the levels of transfected nLuc mRNAs flanked by either GAPDH or CoV2 UTRs (S5C Fig). This is likely due to the lower sensitivity of in-vitro transcribed RNAs to Nsp1-mediated degradation [27] and the short duration of the experiment (4 h post RNA transfection). However, it is also possible that a fraction of transfected mRNAs is retained in endocytic vesicles, which may protect them from Nsp1-induced degradation and potentially confound measurements of intracellular mRNA stability, thus obscuring the destabilizing effects of Nsp1 on host mRNA.

To independently confirm that the stimulatory effect of Nsp1 is indeed the result of increased translation initiation, we repeated the dual transfection experiment and measured whether the presence or absence of Nsp1 affects how much of each nLuc mRNA variant is associated with polysomes. To that end, we performed sucrose gradient ultracentrifugation followed by qPCR of gradient fractions. As expected, Nsp1 expression reduced global translation (Fig 5D, left). Furthermore, it shifted GAPDH-nLuc mRNA from heavy polysomes to lighter fractions containing soluble untranslated mRNAs (Fig 5D, right). In contrast, Nsp1 increased the proportion of CoV2-nLuc mRNA associated with polysomes (Figs 5D and S5D), consistent with specific and selective enhancement of translation initiation on CoV2 gRNA.

## Stimulatory effect of Nsp1 on CoV2 translation is mediated by EIF1/1A and an upstream non-AUG start codon

To better understand the mechanism by which Nsp1 stimulates CoV2 translation, we investigated potential links between Nsp1 and translation factors. Two datasets using proximity

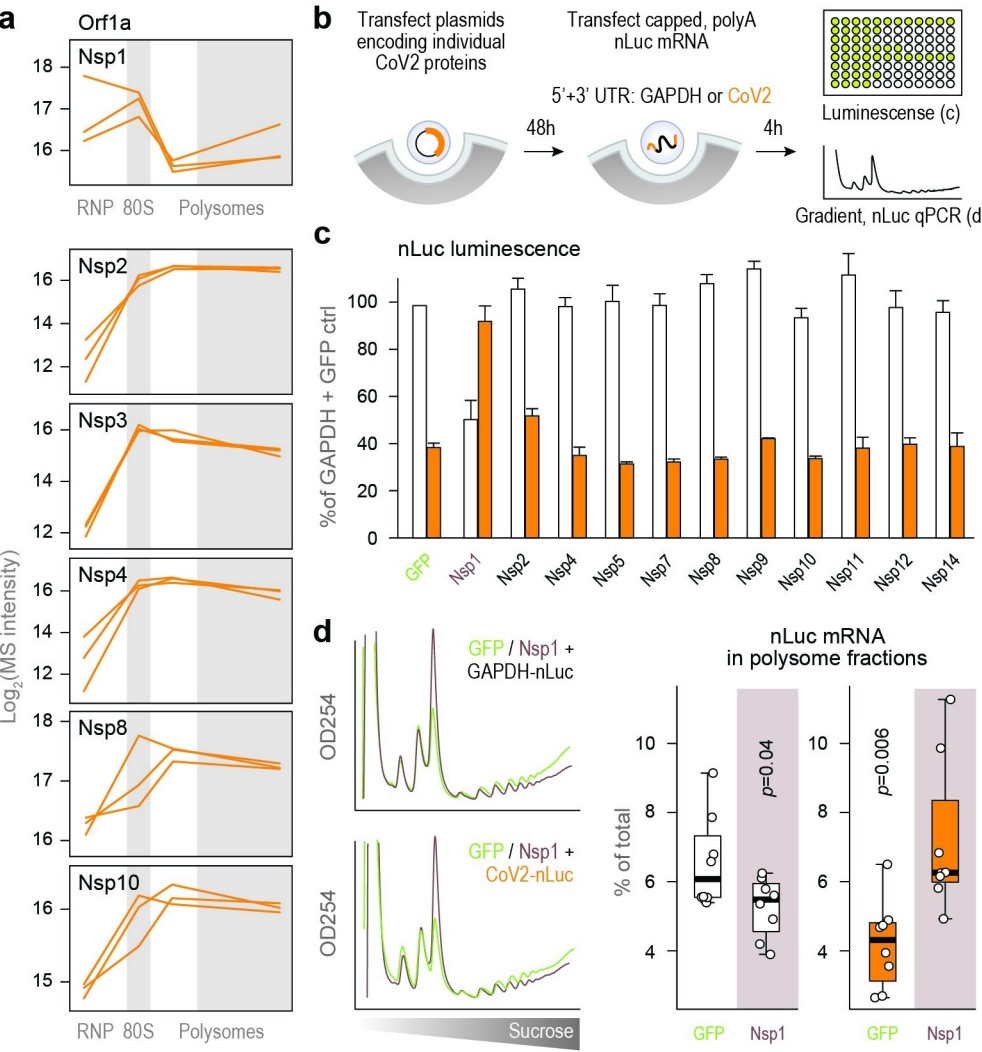

**Fig 5. Nsp1 promotes translation initiation on CoV2 gRNA.** (**a**) Line plots of individual viral proteins quantified by MS in each fraction. Each line represents a single replicate. (**b**) Vero cells were transfected with plasmids encoding for individual CoV2 proteins. At 48 hours post-transfection, cells were transfected with nLuc mRNA flanked by either CoV2 (orange) or GAPDH (white) UTRs. At 4 hours post-second transfection, cells were subjected to either luminescence measurements or sucrose gradients coupled to qPCR of nLuc mRNA. (**c**) nLuc luminescence. Shown are means±SD of 3 independent replicates. (**d**) Cells transfected, as above, with either GFP or Nsp1 followed by GAPDH-nLuc or CoV2-nLuc mRNA, were lysed and fractionated on 10–50% sucrose gradients with continuous monitoring of rRNA. The content of nLuc mRNA in each fraction was determined by qPCR. Left, rRNA absorbance profiles. Right, percent of GAPDH-nLuc or CoV2-nLuc mRNA found in polysome fractions of sucrose gradients. Shown are qPCR measurements of 4 polysome fractions from 2 independent gradients. P, two-tailed Student's t-test p-value.

labeling coupled to proteomic analysis [26,28] found that wild-type Nsp1—but not mutant Nsp1 deficient for ribosome binding—interacts with a subset of translation initiation factors, including EIF1A, EIF3A, DDX3 and EIF4A1. These were also enriched in the heavy polysome fractions of infected cells (Fig 4A). Furthermore, initiation factors EIF1 and 1A were found associated with Nsp1-bound 40S ribosomes in cryo-EM studies [8], although a more recent study reported that Nsp1 may prevent EIF1A binding to ribosomes [29].

Initiation from the correct start codon is critical for translation fidelity, and EIF1 and 1A are important regulators of this process [30–33]. Natural and induced fluctuations in the levels

or activity of these factors affect initiation site stringency [30,34–36], particularly in the context of non-AUG translation of upstream open reading frames (uORFs) [37,38] (reviewed in [39]). Interestingly, CoV2 and other coronaviruses encode for multiple conserved uORFs that are actively translated during infection [21,40]. Ribosome profiling analyses of CoV2-infected cells have shown that ribosomes initiate translation from an upstream CUG codon found at position 59 of all gRNA and sgRNAs, just before the transcription regulatory sequence (TRS) [21,41]. This CUG is one of five codons in the 5'UTR of CoV2 gRNA that can initiate translation [21]. Four of the five potential initiating codons are predicted to translate two short uORFs, both of which terminate ahead of the fifth and main AUG at position 266 (Fig 6A). Template-switching at the TRS generates sgRNAs with conserved CUG but variable downstream sequences, forming uORFs with different initiation contexts that are all—except for Spike sgRNA—out of frame with the main AUG. Given that uORFs are generally thought to repress translation of downstream ORFs [39,42], we speculated that EIF1/1A may affect CoV2 translation by either promoting or inhibiting the bypass of repressive uORFs and therefore control downstream initiation from the main AUG.

To test the role of EIF1/1A in CoV2 infection, we transfected Vero cells with siRNAs, infected with CoV2, and measured virus production by plaque assays. Knockdown of EIF1A, and to a lesser extent, EIF1, reduced CoV2 titers by up to about 10 fold (Fig 6B, left). siRNAs targeting EIF4B did not affect CoV2 titers (Fig 6B, left), suggesting that the effects of EIF1/1A depletion were not simply due to interfering with core components of the protein synthesis machinery. Furthermore, other positive strand RNA viruses e.g. polio (PV), Zika (Zikv) and dengue (DENV) were not affected by knockdown of either EIF1 or 1A (Fig 6B, right), suggesting a CoV2-specific dependency. Next, we performed ribosome profiling analysis of Vero cells transfected as above and infected with CoV2 for 16h. We confirmed efficient knockdown at the protein level (S6A Fig), generated ribo-seq libraries without formaldehyde crosslinking, and aligned ribosome protected fragments to both the virus and the host transcriptome. Reads mapping to CoV2 RNA accounted for about 2–3% of all uniquely aligned reads, regardless of EIF1/1A levels (S6B Fig). Although overall read numbers were similar, their distribution was different. Knockdown of EIF1A, and to a lesser extent, EIF1, led to more ribosomes accumulating on the upstream CUG (Fig 6C and 6D). This was followed by a 3-nt periodicity typical of translated ORFs (S6C Fig), confirming that the CUG codon serves to initiate translation. Under the same conditions, ribosomes spent more time on a downstream in-frame stop codon (S6D Fig) and less time on the two AUG sites at positions 107 (uORF1) and 266 (Orf1), reflected by a respective increase or decrease in ribosome protected reads (Fig 6D). These observations are consistent with an inverse relationship between initiation from the upstream CUG and downstream AUG codons, which is regulated by EIF1/1A.

The increase in ribosome occupancy on CUG(59) affected both gRNA and individual sgRNA, ranging in magnitude from 2 fold (Orf1a, N, M) to 8 fold (Spike, Orf6, Orf3a) upon EIF1A depletion (S6E Fig). EIF1 depletion had similar effects on some but not all viral RNAs (S6E Fig). If CUG-mediated translation of uORF1 inhibits initiation from the main AUG, EIF1/1A depletion should decrease the number of ribosomes on the main ORFs of both gRNA and sgRNAs. Indeed, knockdown of either EIF1 or 1A reduced ribosome occupancy on Orf1a (Fig 6E, left). However, only EIF1A depletion decreased ribosome occupancy on Nucleoprotein (N) and increased ribosome occupancy on Spike coding regions (Fig 6E, middle and right panels). Importantly, these changes in relative translation of gRNA and sgRNAs altered the stoichiometry of CoV2 proteins during infection. Knockdown of EIF1A, and to a lesser extent, EIF1, increased the levels of Spike and decreased the levels of Nsp1 (Fig 6F). Furthermore, knockdown of EIF1A decreased whereas knockdown of EIF1 increased the levels of N protein, consistent with the differences observed by ribosome profiling (compare Fig 6E and 6F).

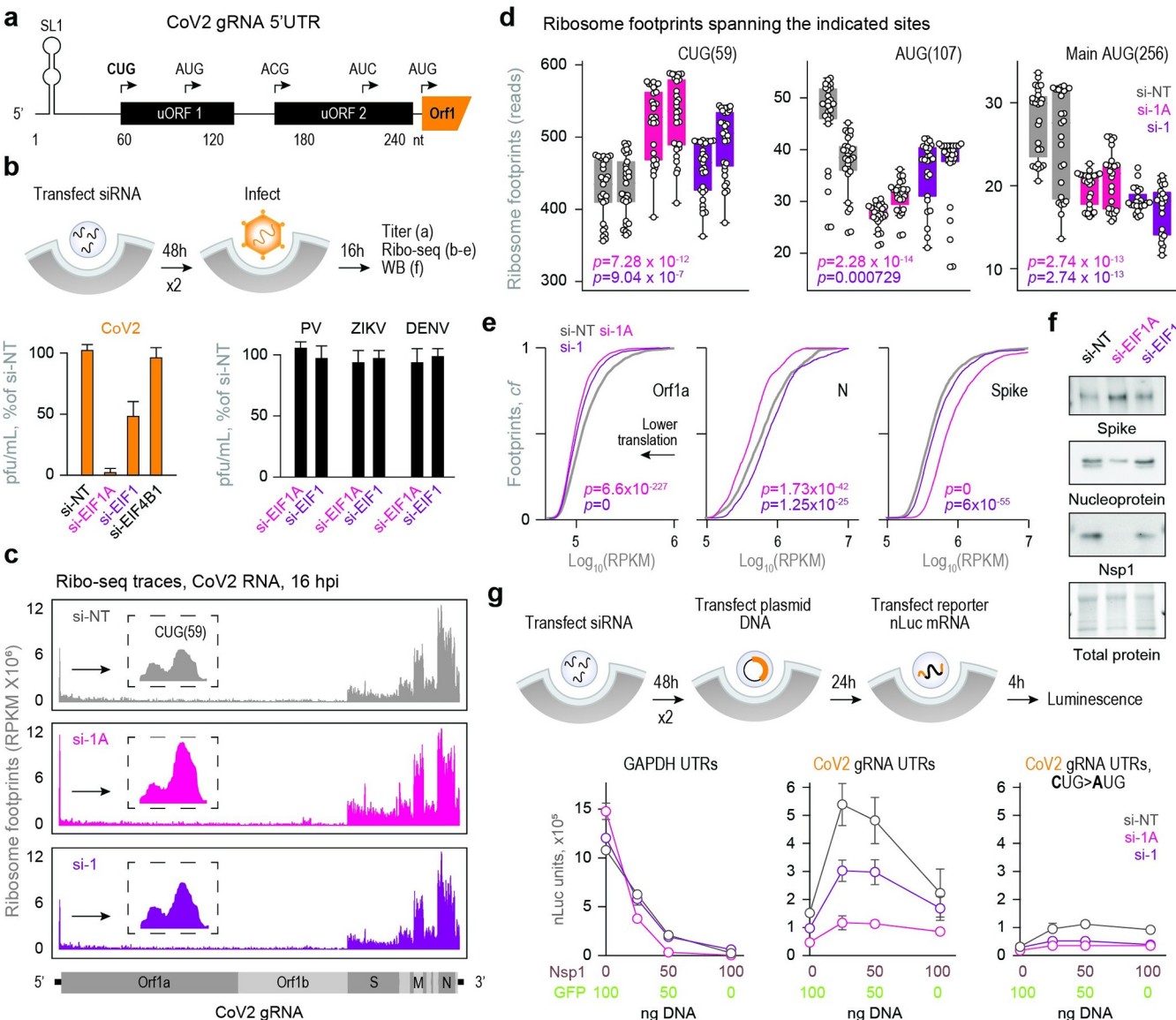

**Fig 6. Nsp1 promotes accurate start codon usage through EIF1/1A.** (**a**) Diagram of elements in CoV2 gRNA 5'UTR involved in translation. SL1, stem-loop 1. uORF1 and uORF2 can both be translated from two different start codons. (**b**) Vero cells were transfected with siRNAs targeting initiation factors 1A, 1, and 4B, compared to non-targeting (NT) controls. At 48h the same transfection was repeated. At 48h after the second transfection, cells were infected with either CoV2 (orange, left) polio (PV), Zika (ZIKV) or dengue (DENV) viruses (black, right) at MOI = 0.5. Viral titers were determined by plaque assays. Shown are means±SD of 4 independent replicates. (**c**) Vero cells were transfected with siRNA as above, infected with CoV2 at MOI = 5, and subjected to ribosome profiling analysis at 16 hpi. Traces show ribosome footprints on CoV2 RNA. Insets, ribosome footprints on nucleotides 20–80 of the gRNA. Representative of 2 independent replicates. (**d**) Ribosome footprints spanning 13 nt upstream and 12 nt downstream of the indicated start codons. Each bar represents a single biological replicate. P, Wilcoxon ranked-sum p-value. (**e**) Cumulative fraction plots of ribosome footprints on Orf1a, Spike and Nucleocapsid (N) open reading frames. Combined analysis of two independent replicates. P, Wilcoxon ranked-sum p-value. A shift of the curve to the left reflects lower ribosome occupancy and therefore lower translation. (**f**) Vero cells transfected with siRNA and infected as in (b) were subjected to immunoblot analysis of whole cell lysates using antibodies against CoV2 Spike, Nucleoprotein and Nsp1. Shown are representative blots of 4 independent repeats. (**g**) Vero cells were transfected with siRNA as above, followed by combined plasmid transfections of GFP and Nsp1 at the indicated amounts. The same total amount of DNA was used for each transfection. At 24 hours, cells were transfected again with nLuc mRNA flanked by UTRs from either GAPDH (left panel), CoV2 gRNA (middle panel) or CoV2 gRNA with CUG(59) mutated to AUG (right panel). At 4 hours post-second transfection, cells were subjected to luminescence measurements. Shown are means±SD of 3 independent replicates.

Such inconsistencies between gRNA and sgRNA, as well as EIF1 and 1A depletion, are likely attributed to the variable initiation context of sgRNAs, conferring different responses to EIF1/1A depletion. Spike sgRNA is unique in that CUG(59) is in frame with the main AUG and therefore promotes—rather than inhibits—translation of the main coding region, particularly during CoV2 infection [41]. Indeed, both EIF1 and 1A knockdown resulted in a proportional increase of ribosome protected reads on both the upstream CUG and the main ORF of Spike gRNA (Figs 6E and S6E).

We next used our reporter assays to test how Nsp1, EIF1 and EIF1A affect initiation from the upstream CUG and downstream AUG. We sequentially transfected Vero cells with targeting or non-targeting siRNAs, GFP or Nsp1 plasmids, and GAPDH- or CoV2-nLuc mRNA. 4 hours after mRNA transfection, we performed RT-qPCR and luminescence measurements. As before, Nsp1 had little effect on mRNA levels for either nLuc reporter (S6F Fig). Consistent with a prior report [9], Nsp1 decreased translation from GAPDH UTRs in a dose-dependent manner, regardless of EIF1/1A levels (Fig 6G, left). In contrast, translation from CoV2 UTRs was stimulated by Nsp1, but only at low to moderate levels (Fig 6G, left). Indeed, Nsp1 was previously shown to stimulate CoV2 translation at low concentrations and inhibit it at high concentrations [26]. The observed stimulation of translation by Nsp1 was dependent on both EIF1 and 1A, because depletion of EIF1 lowered and 1A nearly abrogated the Nsp1-mediated increase in translation (Fig 6G, middle panel). Finally, replacing CUG(59) with AUG to enforce higher translation [43] of uORF1 strongly inhibited translation of the main ORF. Under these conditions, Nsp1 failed to stimulate translation at any of the concentrations tested (Fig 6G, right).

Taken together, these observations imply that Nsp1 may act as a molecular switch balancing translation of gRNA and sgRNA during infection by influencing initiator codon choice. To explore this hypothesis, we reanalyzed ribosome profiling datasets generated from Calu3 cells infected with either WT or Nsp1-dead CoV2 [22]. This mutant, in which amino acids 155–165 of Nsp1 were deleted to disrupt its ribosome binding, is unable to suppress host translation and shows delayed replication kinetics [22]. At 4 hpi, loss of Nsp1 function reduced translation of gRNA and sgRNA by up to about 95 and 90%, respectively (S6G Fig). However, at 7 hpi, translation of most sgRNA recovered to nearly WT levels whereas that of gRNA barely increased at all (S6G and S6H Fig). Our analyses thus indicate that early accumulation of Nsp1 preferentially stimulates translation of CoV2 gRNA. Indeed, for WT-CoV2, translation efficiency of gRNA increased from 4 to 7 hpi, but this increase was not observed with Nsp1-dead CoV2 (S6I Fig).

Finally, it was previously proposed that certain cellular mRNAs may also escape translation repression by Nsp1 [44]. Therefore, we wondered whether the presence of uORFs in cellular mRNAs may confer protection from Nsp1 repression. Notably, we find that Nsp1 promotes uORF bypassing not only on viral but also cellular mRNAs. At 4 hpi, translation of uORF-repressed host transcripts was higher in the presence of Nsp1, and this difference was proportional to the number of uORFs in each 5'UTR (S6J Fig).

## Discussion

Despite intense research efforts, much is still unknown about the complex interactions between CoV2 and its host cell. Protein-protein and protein-RNA interactions have been characterized using various methods [3,12,18,45], but their functional significance remains poorly understood, and more work is needed to determine whether they reflect druggable viral dependencies. Several lines of evidence suggest that viral protein synthesis involves unique challenges and vulnerabilities that may be exploited for the development of host-directed

antivirals. Viral RNAs are highly structured [45] and harbor multiple overlapping open reading frames [21], two of which encode for long multifunctional polyproteins. The longest, Orf1ab, is synthesized as a single protein of 7,096 amino acids—more than 10 times the average length of a human protein [46]—and then processed into 16 individual subunits, each with its own unique structure, function, and interaction networks [3,47]. The recruitment of diverse protein synthesis, folding and degradation factors to CoV2 polysomes, reported in this study, as well as the identification of translation inhibitors with selective antiviral effects [3,4], further illustrate the complexity of biosynthetic dependencies between CoV2 and its host.

Our understanding of CoV2 translation efficiency remains limited. Some reporter assays suggest that translation of all CoV2 RNA species is highly efficient, or at a minimum similar to that of cellular mRNAs encoding housekeeping proteins [9,48,49]. However, a detailed mutagenesis analysis of CoV2 5'UTR identified multiple sequence and structure elements that repress translation initiation from the main AUG [50]. In addition, ribosome profiling and other sequencing and immunofluorescence studies have argued that viral translation efficiency is lower than that of cellular mRNAs [6,22], although sequencing-based measurements may be biased by the presence of positive-strand viral RNA in replication, transcription and packaging complexes. Such discrepancies may also be explained by differences in experimental design, including variability in translation components between in-vitro translation lysates and intact cells, as well as the different sensitivity to Nsp1 exhibited by mRNAs transcribed in-vitro and in-cellulo [27].

Measurements of CoV2 translation efficiency are further complicated by Nsp1, which inhibits translation initiation and promotes degradation of cellular but not CoV2 RNA [6–9,22,51]. A conserved stem-loop (SL1) found at the 5' end of all viral RNAs (Fig 6A) protects them from Nsp1-induced repression [7,48], though it remains unclear whether such protective effects are mediated by direct interactions that displace Nsp1 from ribosomes. At low concentrations, Nsp1 was even found to enhance translation of reporter mRNAs harboring either the full-length 5'UTR or SL1 only [26]. In our hands, Nsp1 suppressed translation from GAPDH UTRs and promoted that of CoV2 gRNA UTRs, but only at low to intermediate concentrations (Fig 6G). Therefore, variability in the intracellular levels of Nsp1 across cell types, transfection methods and other experimental conditions is an additional confounding factor when measuring CoV2 translation.

We report here, for the first time, that stimulation of CoV2 translation by Nsp1 depends on initiation factors EIF1 and 1A, which were enriched in heavy polysome fractions of infected cells (Fig 4A). EIF1 and 1A stabilize the open conformation of the pre-initiation complex, allowing the initiator tRNA to inspect the mRNA sequence for complementarity to the anticodon [31,32]. EIF1 discriminates against suboptimal initiation sites [38] to prevent excessive uORF translation genome-wide [37], likely by preventing the GTP hydrolysis that commits the ribosome to initiation at a specific codon. EIF1A can either promote or inhibit such commitment, through the competing functions of its two termini (reviewed in [33]). The nature of interactions between Nsp1 and EIF1/1A remains elusive, with some reports detecting them on Nsp1-bound ribosomes [8,26,28] while others suggesting they are displaced from ribosomes by Nsp1 [9,29]. However, our experiments reveal a clear functional link between Nsp1 and EIF1/1A, since knockdown of either EIF1 or 1A altered Nsp1-dependent functions. It increased ribosome accumulation on CUG(59) (Fig 6D); lowered the stimulatory effect of Nsp1 (Fig 6G); altered the relative translation of gRNA and individual sgRNA during infection (Fig 6E); and disrupted the stoichiometry of viral proteins (Fig 6F). Interestingly, a recent study found that depletion of EIF1A activates NF-κB and interferon gamma signaling [52]. It is tempting to speculate that host cells have evolved mechanisms to sense the hijacking of EIF1A by viruses and counter it by triggering an innate immune response.

Other translation factors and RNA helicases implicated in start site selection were also recruited to heavy polysomes during infection (Fig 4A). For example, EIF3A regulates re-initiation after termination of uORF translation [53,54], and DDX3 facilitates translation of mRNAs with complex 5'UTR [55] and promotes CoV2 infection [56]. Many RNA helicases are also known to bind CoV2 RNA [12,57] and modulate infection [58]. Future work will need to determine how EIF1/1A depletion remodels polysome interactions during CoV2 infection, and whether additional factors contribute to initiation fidelity on CoV2 RNAs.

uORFs are common translation repressors of cellular mRNAs [42], shaping the proteome during stress [59]. Our work sheds light on uORFs acting as conserved translation regulatory elements found in the 5'UTR of CoV2 and other viruses. uORFs initiating from either AUG or non-AUG codons are found in various coronaviruses [21,40,60]. For example, uORF1 in mouse hepatitis virus (MHV) is initiated from either a UUG codon located just upstream of the TRS or a more downstream in-frame AUG codon [40], similar to CoV2 (Fig 6A). MHV uORF1 suppresses translation of Orf1, but plays a nonessential role in viral replication in cell cultures. However, mutations disrupting translation initiation of uORF1 are quickly reversed upon passaging [60]. Similarly, a conserved uORF in enteroviruses is dispensable for replication in immortalized cultures but promotes infection in human intestinal organoids [61]. These observations suggest that the conserved presence of multiple translation repressors in the 5'UTR of viruses may serve pro-viral functions, at least under certain conditions. This might explain the discrepancy between our current work and previous reports showing that SL1 alone, without the downstream CUG, is sufficient to confer a translation boost by Nsp1 [26]. In contrast to SL1-only reporter mRNAs, all viral RNAs generated during CoV2 infection harbor repressive uORF(s) that must be bypassed to allow efficient synthesis of viral proteins. Furthermore, our CoV2 reporters escaped repression by Nsp1 at all concentrations and regardless of CUG(59), EIF1 or EIF1A (Fig 6G), indicating that escape and stimulation may occur through distinct mechanisms. Our work thus indicates that a complex interplay between viral Nsp1 and host initiation factors are exploited by the virus to regulate initiation of its various ORFs and control the timing and stoichiometry of viral protein synthesis. Although it remains unclear whether Nsp1 can regulate translation start site selection independently of EIF1/1A, this dependency may offer potential new targets for antiviral therapies.

## Materials and methods

### Cell cultures and viral infection

The African green monkey kidney Vero E6 (ATCC, no. CRL-1586) and human Calu3 cells (ATCC, no. HTB-55) were grown in DMEM medium (Thermo Fisher) supplemented with 10% fetal bovine serum, 100 units/ml penicillin and 100 mg/ml streptomycin, at 37˚C in a 5% $CO_2$ incubator. A clinical isolate of SARS-CoV-2 (USA-WA1/2020, BEI Cat No: NR-52281) was propagated in Vero cells and was used for Vero experiments. Viral titers were quantified with a plaque assay. All infections were performed at biosafety level-3 (BSL-3). To assess antiviral activity, ~70% confluent monolayers of Vero cells ($3 \times 10^5$ cells/well in 24-well plates) were pretreated with drugs or drug combinations at indicated concentrations for 3 hours (pretreatment) and then infected with SARS-CoV-2 (MOI = 0.5) at 37˚C for 1 hour. The virus solution was removed, cells were further cultured with fresh medium containing drugs at the same concentrations. At 16 hours post-infection, supernatants were collected, and viral titers were measured with a plaque assay. Type I poliovirus (Mahoney), type 2 dengue (Thailand/16681/84) and Zika virus (PRVABC59) were generated, passaged and tittered as described [5], at biosafety level-2 (BSL-2).

## Plaque assay

Confluent monolayers of Vero cells grown in six-well plates were incubated with the serial dilutions of virus samples (250 μl/well) at 37˚C for 1 hour. Next, the cells were overlayed with 1% agarose (Invitrogen) prepared with MEM supplemented with 2% FBS and antibiotics. Three days later, cells were fixed with 4% formaldehyde for 2 hours, the overlay was discarded, and samples were stained with crystal violet dye, and the number of plaques was calculated.

## Polysome profiles

A total of $1-5 \times 10^7$ Vero or Calu3 cells in T175 flasks were washed twice with and scraped into ice-cold PBS with calcium and magnesium. Cells were pelleted at 1,200 RCF for 5 min at 4˚C, and supernatants were removed. Pellets were resuspended in 220 μl polysome buffer (25 mM HEPES pH = 7.5, 100 mM NaCl, 10 mM $MgCl_2$, 2 mM dithiothreitol (DTT) and Complete EDTA-free protease inhibitor cocktail (Millipore Sigma). Triton X-100 and sodium deoxycholate were added to a final concentration of 1% each and the samples were incubated on ice for 20 min and centrifuged at 20,000 RCF for 10 min at 4˚C to remove cell debris. 250 μl lysates were transferred to fresh tubes, combined with 125 μl 12% formaldehyde, and incubated on ice for 30 min. To quench the crosslinking reaction, samples were incubated with 125 μl 4M Tris pH = 8.0 on ice for 20 min, followed by flash freezing. Frozen lysates were thawed on ice and loaded on 10–50% sucrose gradients in Polysome buffer and subjected to ultracentrifugation at 36,000 rpm in an SW41.Ti swinging bucket rotor (Beckman Coulter) for 150 min at 4˚C. 15 Equal volume fractions were collected using Gradient Station (BioComp) with continuous monitoring of rRNA at UV254. Fractions were pooled as follows: 1–4 for free RNP; 5 for 80S monosomes; 6–8 for light polysomes, and 9–15 for heavy polysomes. For polysome profiles of cells transfected with nanoluciferase reporters, the above protocol was followed without formaldehyde fixation or pooling of gradient fractions.

## Sample preparation for proteomic analysis

Pooled gradient fractions were diluted 1:2 with ice-cold PBS and tumbled overnight at 4˚C with 50 μl Strataclean resin (Agilent). Beads were pelleted at 600 RCF for 5 min at 4˚C, and supernatant was removed. Beads were resuspended in 100 μl PBS supplemented with 2% SDS and incubated at 95˚C for 15 min to elute proteins from beads and reverse formaldehyde crosslinks. Beads were pelleted at 600 RCF for 5 min at room temperature, and supernatants were transferred to fresh tubes. Protein was extracted using methanol-chloroform precipitation: 400 μl methanol, 100 μl chloroform and 350 μl water were added sequentially to each 100 μl sample, followed by centrifugation at 14,000 RCF for 5 min at room temperature. The top phase was removed, and the protein interphase was precipitated by addition of 400 μl methanol, followed by centrifugation at 14,000 g for 5 min at room temperature. Pellets were air dried and resuspended in 8M urea, 25 mM ammonium bicarbonate (pH 7.5). Protein concentration was determined by BCA (Thermo Fisher) and 1–2 μg total protein were subjected to reduction and alkylation by incubation with 10 mM DTT for 1 h at room temperature followed by 5 mM iodoacetamide for 45 min at room temperature, in the dark. The samples were then incubated with 1:50 enzyme to protein ratio of sequencing-grade trypsin (Promega) overnight at 37˚C. Peptides were desalted with μC18 Ziptips (Millipore Sigma), dried and resuspended in 10 μL 0.1% formic acid in water.

## LC-MS/MS acquisition

Digested peptides were resuspended in 0.1% formic acid and 40% of each specimen was analyzed on an LTQ Orbitrap Fusion Lumos Tribrid Mass Spectrometer (Thermo Fisher

Scientific) coupled with a Dionex Ultimate 3000 liquid chromatography system (Thermo Fisher Scientific). Peptides were separated by capillary reverse-phase chromatography for 120 min on a 24-cm reversed-phase column (inner diameter of 100 μm, packed in-house with ReproSil-Pur C18-AQ 3.0 m resin (Dr. Maisch)). A multi-step linear gradient was applied as follows: 96% A + 4% B to 75% A + 25% B over 70 min; 75% A + 25% B to 60% A + 40% B over 20 min; 60% A + 40% B to 2% A + 98% B over 2 min and maintain this proportion for 2 min before returning to 98% A + 2% B and holding this proportion for 24 min. Buffer A is 0.1% formic acid in water and buffer B is 0.1% formic acid in acetonitrile; flow rates were maintained at 300 nl/min throughout the gradient. Full MS scans of intact peptide precursor ions were acquired in the Orbitrap mass analyser at a resolution of 120,000 (FWHM) and m/z scan ranges of 400–1,500 in a data-dependent mode. The automatic gain control (AGC) target was $4 \times 10^5$, the maximum injection time was 50 ms, and the isolation width was 1.6 Da. The most intense ions recorded in full MS scans were then selected for MS2 fragmentation in the Orbitrap mass analyser using higher-energy collisional dissociation (HCD) with a normalized collision energy of 30% and resolution of 15,000 (FWHM). Monoisotopic precursor selection was enabled, and singly charged ion species and ions with no unassigned charge states were excluded from MS2 analysis. Dynamic exclusion was enabled, preventing repeated MS2 acquisitions of precursor ions within a 10 ppm m/z window for 15 s. AGC targets were $5 \times 10^4$ and the maximum injection time was 100 ms.

## Mass spectrometry data processing

Raw MS data were processed using MaxQuant version 1.6.7.0 [62]. MS/MS spectra searches were performed using the Andromeda search engine [63] against the forward and reverse human and mouse Uniprot databases (downloaded August 28, 2017 and November 25, 2020, respectively). Cysteine carbamidomethylation was chosen as fixed modification and methionine oxidation and N-terminal acetylation as variable modifications. Parent peptides and fragment ions were searched with maximal mass deviation of 6 and 20 ppm, respectively. Mass recalibration was performed with a window of 20 ppm. Maximum allowed false discovery rate (FDR) was <0.01 at both the peptide and protein levels, based on a standard target-decoy database approach. The "calculate peak properties" and "match between runs" options were enabled. All statistical tests were performed with Perseus version 2.0.7.0 using either ProteinGroups or Peptides output tables from MaxQuant. Potential contaminants, proteins identified in the reverse dataset and proteins only identified by site were filtered out. For proteins not annotated in the Chlorocebus sabaeus proteome, we manually added the human ortholog gene symbol where similarity was >90%. Human ortholog Uniprot IDs were added based on gene symbol. Intensity-based absolute quantification (iBAQ) was used to estimate absolute protein abundance. Data was log2 transformed and median adjusted. Two-sided Student's t-test with a permutation-based FDR of 0.01 and S0 of 0.1 with 250 randomizations was used to determine statistically significant differences between grouped replicates. Categorical annotation was based on Gene Ontology Biological Process (GOBP), Molecular Function (GOMF) and Cellular Component (GOCC), as well as protein complex assembly by CORUM.

## Drug treatments, toxicity and synergy

To determine the effects of drugs on host cells, $5\times10^4$ Vero cells were seeded per well in a black wall, clear bottom 96-well plate. Drugs were added 24 h after seeding. 24 h after adding drugs, cell viability was assayed using CellTiter-Glo (Promega), according to the manufacturer's instructions, on a Tecan Ultra Evolution microplate reader. To determine the effects of drugs on CoV2 infection, virus adsorption was performed as described above, at MOI = 0.5, and

inoculum was replaced with fresh media containing drugs. Cell supernatants were collected at 16 hpi and subjected to plaque assays. SynergyFinder2 [64] was used to calculate Bliss scores.

## Nanoluciferase reporters, viral proteins, and RNA/DNA transfections

Plasmids encoding for nanoluciferase (nLuc) flanked by 5' and 3' untranslated regions of either CoV2 or GAPDH, and plasmids encoding for strep-tagged individual CoV2 proteins were kind gifts from Joseph (Jody) Puglisi [9] and Nevan Krogan [3]. To generate translation-competent mRNA, nLuc plasmids were linearized using SpeI and subjected to in-vitro transcription using HiScribe T7 kit (New England Biolabs), according to the manufacturer's instructions. In 20 μL total volume per reaction, 1 μg plasmid was combined with 2 μL buffer, 2 μL of each nucleotide, 1.6 μL cap, 1 μL SUPERase-in and 2 μL enzyme. Transcription was performed for 3 h at 37°C and terminated by precipitation. Each sample was combined with 1:1 v/v 5 M ammonium acetate, incubated on ice for 15 min, and centrifuged 20,000 RCF for 30 min at 4°C. Pellets were washed with 1 mL 75% ethanol, air dried and resuspended in 50 μL nuclease-free water. RNA integrity was confirmed using agarose gels stained with SYBR Safe, and concentration was determined by nanodrop.

For RNA transfections, Vero cells were plated at $2x10^4$ cells/well in 96-well plates. The following day, transfection reactions were prepared as follows (per well). 0.15 μL Lipofectamine MessengerMAX (Thermo) was diluted in 5 μL OptiMEM (Thermo) and incubated for 10 min at RT. 20 ng in-vitro transcribed RNA was diluted in 5 μL OptiMEM and mixed with the MessengerMAX solution for 5 min. 10 μL transfection reactions were added per well, and cells were harvested at 4 h post-transfection. Luminescence measurements were acquired using Nano-Glo Luciferase Assay System (Promega), according to the manufacturer's instructions, on a Tecan Ultra Evolution microplate reader.

For DNA/RNA transfections, Vero cells were plated at $2x10^4$ cells/well in 96-well plates. The following day, transfection reactions were prepared as follows (per well). 1 μL Lipofectamine 2000 (Thermo) was diluted in 5 μL OptiMEM and 25 ng plasmid DNA was diluted in 5 μL OptiMEM. The two solutions were mixed, incubated at RT for 5 min, and added to cells. Media was replaced at 6 h post-transfection. RNA transfections were performed as above, 24 h after DNA transfections. For DNA/RNA transfections prior to sucrose gradient fraction, the above procedure was scaled up 500x, and cells were collected for polysome profiled analysis, as described above.

## Quantitative Real-Time PCR (qRT-PCR)

Total RNA was extracted using Trizol (Invitrogen) according to the manufacturer's instructions. To extract RNA from sucrose gradient fractions, 1 μl pellet paint co-precipitant (Millipore Sigma) and 500 μl phenol:chloroform:isoamyl alcohol (25:24:1) were added to each 500 μl fraction and incubated for 5 min at room temperature. Phase separation was performed at 12,000 RCF for 15 min at 4°C, and the top phase was removed and subjected to another round of extraction as above. 400 μl of the top phase was combined with 600 μl isopropanol and RNA was pelleted at 12,000 RCF for 1 h at 4°C. Pellets were washed with 1 ml 75% ice-cold ethanol, air dried and resuspend in 20 μl RNase-free water. cDNA was synthesized using the High Capacity cDNA Reverse Transcription Kit (Thermo Fisher), according to the manufacturer's instructions, using 5 μl of RNA from each gradient fraction. qRT-PCR analysis was performed using SensiFast SYBR (BioLine) and gene-specific primers (nLuc FWD 5' CAGCCGGCTACAACCTGGAC 3', REV 5' AGCCCATTTTCACCGCTCAG 3'; GAPDH FWD 5' AGGTCGGAGTCAACGGAT 3', REV 5' TCCTGGAAGATGGTGATG 3'; ACTB FWD 5' GGCTGTATTCCCCTCCATCG 3', REV 5' CCAGTTGGTAACAATGCCATGT 3';

18S rRNA FWD 5′ `GTAACCCGTTGAACCCCATT` 3′, REV 5′
`CCATCCAATCGGTAGTAGCG` 3′), according to the manufacturer's instructions. To estimate
relative abundance of specific mRNAs in each gradient fraction, each Ct value was divided by
the sum of Ct values across all gradient fractions.

## siRNA transfections

siRNAs against EIF1AX, EIF1 and EIF4B (Smartpool M-011262-02-0005, M-015804-01-0005,
L-020179-00-0005), as well as nontargeting siRNAs (Smartpool D-001206-13-05), were from
Dharmacon. siRNAs were reconstituted in nuclease-free water to a concentration of 10 μM.
Vero cells were plated at $1x10^5$ cells/well of 12-well plates. The following day, 3 μl RNAiMax
reagent (Thermo) was diluted in 50 μl OptiMEM and combined with 2 μl siRNA in 50 μl Opti-
MEM. After 5 min at RT, the reaction mix was added to cells, and media was replaced after 6
h. At 48 h post-transfection, cells were passaged and transfected again, as above. Additional
DNA/RNA transfections were launched 48 h after the second siRNA transfection.

## SDS-PAGE and immunoblotting

For immunoblotting, cells were washed twice with ice-cold PBS and lysed on plate with RIPA
buffer (25 mM Tris-HCl pH = 7.5, 150 mM NaCl, 1% NP-40, 0.5% Sodium deoxycholate) sup-
plemented with 2 mM DTT, Complete EDTA-free protease inhibitor cocktail, and 50 units/
mL benzonase (Millipore Sigma) to remove DNA. Lysis was performed on ice for 20 min and
lysates were clarified by centrifugation for 10 min at 12,000 RCF at 4˚C. Protein concentration
was determined by BCA assay (Thermo Fisher) and 4x Laemmli sample buffer (Bio-Rad) sup-
plemented with fresh 10% 2-mercaptoethanol was added to a final concentration of x1. 15 μg
of each sample was resolved on 4–20% SDS-PAGE (Bio-Rad), transferred to 0.2 μm PVDF
membranes presoaked in methanol for 30 sec. Membranes were blocked with 4% molecular
biology grade BSA (Millipore Sigma) in tris-buffered saline supplemented with 0.1% Tween-
20 (Millipore Sigma) (TBST) for 1 h at room temperature then probed with specific primary
antibodies for 2 h at room temperature. Primary antibodies were diluted in 4% BSA/TBST
supplemented with 0.02% sodium azide, as follows: rabbit anti-EIF1A (1:1000, GeneTex
GTX118810), rabbit anti-EIF1 (1:1000, ProteinTech 15276-1-AP), mouse anti-beta tubulin
(1:10,000, EMD 05–661). Secondary antibodies were diluted 1:10,000 in TBST. Western blot
detection was done using ECL Plus Western Blotting Substrate (Thermo Fisher) and images
were taken either by film radiography.

## Ribosome profiling analysis

Ribosome footprints were prepared essentially as described [65]. EIF1A and nontargeting
siRNA were transfected into Vero cells as above, and 24 h after the second transfection, cells
were replated at $2x10^6$ per T25 flask. CoV2 infection was launched the following day at
MOI = 5. At 16 hpi, cells were washed with 5 mL ice-cold PBS with calcium and magnesium.
PBS was fully aspirated and replaced with 350 μl polysome buffer (25 mM HEPES pH = 7.5,
100 mM NaCl, 10 mM $MgCl_2$, 2 mM DTT and Complete EDTA-free protease inhibitor cock-
tail) supplemented with 1% Triton X-100 and 1% sodium deoxycholate. On-plate lysis was
allowed to proceed on ice for 10 min with occasional shaking. Lysates were transferred to fresh
tubes and clarified at 12,000 RCF for 5 min at 4˚C. 200 μl clarified lysate were combined with
5 μl RNase I (10 U/μl, Invitrogen). Reactions were incubated at RT for 45 min with shaking,
and terminated by addition of 200 U Superase-In (Thermo). Spin columns were used to sepa-
rate monosome, as follows. S-400 HR MicroSpin Columns (Amersham) were drained at 600
RCF for 4 min at 4˚C and the flowthrough was discarded. Resin was resuspended in 200 μl

polysome buffer and centrifuged again. Each 200 μl RNase-digested sample was loaded on 2 separate columns (100 μl each), centrifuged at 600 RCF for 2 min at 4°C. The flowthroughs were combined and 800 μl Trizol (Invitrogen) was added. RNA was extracted according to the manufacturer's instructions. Footprints were resolved on 8M urea 15% PAGE. 15–35 nt fragments were extracted, dephosphorylated, linker ligated according to the protocol. rRNA depletion was performed using Ribo-Zero rRNA Removal Kit (Illumina), according to the manufacturer's instructions. All downstream steps were performed as described [65]. Libraries from two independent repeats were sequenced on a NextSeq 2000 P3 (Illumina). After demultiplexing, sequencing reads were trimmed of adaptor sequences and quality filtered using cutadapt (-a CTGTAGGCACCATCAAT -m1 -q20). Ribosomal RNA was removed using Bowtie2 (—un). Remaining reads 28–32 nt in length were aligned to SARS-Cov-2 genome (Genebank MN985325) using Hisat2 (—trim5 1). Per-nucleotide ribosome occupancy tables were generated using bamCoverage and visualized on Integrated Genome Viewer. For periodicity analysis, a p-site offset of 13 nt was used. For re-analysis of previously published ribosome profiling datasets, read count tables for both mRNA and ribosome footprints were downloaded from GSE200422 [22]. For uORF annotation, we used a previous dataset generated using ribosome profiling (dataset EV3) [42].

## Supporting information

**S1 Fig. Subcellular proteomics of CoV2 infected cells.** (**a**) Calu3 cells were infected with SARS-CoV-2 USA/WA1/2020 at MOI = 5, lysed, fixed with formaldehyde and fractionated on 10–50% sucrose gradients with continuous monitoring of rRNA absorbance. Each line reflects a single replicate, and bar graphs show the ratio of polysomes to sub-polysomes, calculated as the area under the curve (AUC) of relevant fractions. Shown are means±SD of 4 independent replicates. (**b**) Timecourse of CoV2 RNA translation and intracellular viral protein accumulation, from [10]. Shown are means±SD of 3 independent replicates. (**c**) Total protein extracted from pooled fractions of infected and uninfected cells, quantified by bicinchoninic acid (BCA) assays. Each line reflects a single replicate. (**d**) Boxplots of all cytosolic ribosomal proteins quantified by MS in each pooled fraction from all 3 replicates. (**e**) Line plots of RPS6 and CoV2 Nsp12 (RdRp) quantified by MS in each pooled fraction. Each line represents a single replicate.
(TIF)

**S2 Fig. Proteostasis factors recruited to heavy polysome fractions upon CoV2 infection.** Line plots of individual ribosomal proteins quantified by MS in each pooled fraction. Each line represents a single replicate. P, two-tailed Student's t-test p-value of differences in indicated protein abundance in heavy polysome fractions.
(TIF)

**S3 Fig. Toxicity and antiviral effects of proteostasis modulators.** (**a-b**) Vero cells were infected with CoV2 at MOI = 0.5. Single drugs (a) or drug combinations (b) were added at the start of infection, and titers were determined by plaque assays at 16 hours post-infection. Toxicity was determined using CellTiter-Glo at 24h of drug treatment, in the absence of CoV2 infection. Shown are means±SD of 3 independent replicates, normalized to DMSO controls.
(TIF)

**S4 Fig. Inefficient translation initiation on CoV2 gRNA.** (**a**) Change in abundance of individual translation initiation factors upon CoV2 infection, in each fraction. (**b**) Line plots of individual translation factors quantified by MS in each fraction. Each line represents a single replicate. P, two-tailed Student's t-test p-value. (**c**) Translation initiation factors are highly

represented in the CoV2 RNA interactome during infection. Shown are pairwise comparisons of individual host protein abundance, quantified by MS, that specifically interact with either genomic or subgenomic CoV2 RNA. Inset, cumulative distribution plots of 40S and 60S ribosomal protein interaction with CoV2 RNA. (**d**) rRNA absorbance profiles showing lower abundance of free 40S subunits during CoV2 infection. Sum of four replicates.
(TIF)

**S5 Fig. Sucrose sedimentation patterns for individual CoV2 proteins.** (**a**) Line plots of individual viral proteins quantified by MS in each fraction. Each line represents a single replicate. (**b**) Vero cells were transfected with either GFP or NSP1. At 24h, intracellular RNA was extracted and subjected to qPCR analysis using primers specific to the coding region of either GAPDH or ACTB, as well as 18S rRNA. Shown are means±SD of 2 independent replicates, normalized to rRNA levels. (**c**) Vero cells were transfected with either GFP or Nsp1. At 24h, cells were transfected again with GAPDH-nLuc or CoV2-nLuc mRNA. 4h post second transfection, intracellular RNA was extracted and subjected to qPCR analysis using primers specific to the coding region of nLuc. Shown are means±SD of 2 independent replicates, normalized to rRNA levels. (**d**) Cells transfected as above with either GFP or Nsp1 followed by GAPDH-nLuc or CoV2-nLuc mRNA were lysed and fractionated on 10–50% sucrose gradients with continuous monitoring of rRNA. The content of nLuc mRNA in each gradient fraction was determined by qPCR using primers specific to the coding region of nLuc. Gradient qPCR values were calculated as the relative proportion of nLuc mRNA in each fraction compared to the sum of all fractions. Shown are means±SD of 2 independent replicates.
(TIF)

**S6 Fig. Effects of Nsp1, EIF1A and EIF1 on translation.** (**a**) Vero cells were transfected with siRNAs targeting initiation factors 1A and 1, compared to non-targeting (NT) controls. At 48h the same transfection was repeated. At 48h after the second transfection, cells were subjected to immunoblot analysis of whole cell lysates using the indicated antibodies. Shown are representative blots of 2 independent repeats. (**b**) Read mapping statistics for the ribosome profiling analyses in Fig 6B. Each bar represents a single biological replicate. (**c**) Periodicity analysis is consistent with translation of CoV2 gRNA uORF1 from CUG(59). P-site offsets were 13 nt from the 5' end of each read. Shown is data from infection of control si-NT cells. (**d**) Heatmap showing log2 differences in ribosome occupancy on CoV2 gRNA 5'UTR between cells transfected with siRNAs against EIF1A and EIF1 as compared to non-targeting controls. (**e**) Footprints spanning CUG(59), assigned to gRNA (Orf1a) or individual sgRNA. (f) qPCR analysis of intracellular nLuc reporter mRNA in cells transfected with either GFP or Nsp1 followed by nLuc mRNA flanked by the indicated UTRs. Values were normalized to 18S rRNA. Shown are means±SD of 2 independent replicates. (**g-i**) Reanalysis of ribosome profiling datasets generated from Calu3 cells infected with either WT or Nsp1-dead CoV2. (g) Percent difference in footprints mapping to the indicated ORFs between WT-CoV2 and Nsp1-dead CoV2 at 4 hpi (light orange) and 7 hpi (dark orange). (h) Footprints mapping to Orf1a, N or S coding regions at 7 hpi. (i) Translation efficiencies of Orf1a, N and S, calculated as the ratio of ribosome protected reads to mRNA. Shown are means±SD of 2 independent replicates. (**j**) Boxplots showing the difference in translation of host mRNAs during infection with either WT or Nsp1-dead CoV2, binned by number of annotated uORFs. P, p-value of Mann-Whitney test comparing 0 and >5 uORFs.
(TIF)

**S1 Table. Mass-spectrometry data, MaxQuant ProteinGroups output.**
(XLSX)

**S2 Table. Gene Ontology (GO) Fisher enrichment analysis.**
(XLSX)

## Acknowledgments

We thank Angela Detweiler, Sheryl Paul, Honey Mekonen and Norma Neff of the Chan Zuckerberg Biohub San Francisco for their help with sequencing the ribo-seq libraries.

## Author Contributions

**Conceptualization:** Ranen Aviner, Judith Frydman, Raul Andino.

**Data curation:** Ranen Aviner, Yinghong Xiao, Michel Tassetto, Damian Kim, Patrick L. McAlpine, Joshua Elias, Judith Frydman.

**Formal analysis:** Ranen Aviner, Peter V. Lidsky, Yinghong Xiao, Michel Tassetto, Joshua Elias, Judith Frydman, Raul Andino.

**Funding acquisition:** Judith Frydman, Raul Andino.

**Investigation:** Ranen Aviner, Peter V. Lidsky, Yinghong Xiao, Michel Tassetto, Damian Kim, Lichao Zhang, Patrick L. McAlpine.

**Methodology:** Damian Kim, Lichao Zhang, Patrick L. McAlpine, Joshua Elias.

**Resources:** Lichao Zhang, Joshua Elias.

**Supervision:** Judith Frydman.

**Visualization:** Ranen Aviner, Raul Andino.

**Writing – original draft:** Ranen Aviner, Peter V. Lidsky, Judith Frydman, Raul Andino.

**Writing – review & editing:** Ranen Aviner.

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
