## [Decision Letter · Decision Letter 0]

18 Aug 2023

Dear Dr. Andino,

Thank you very much for submitting your manuscript "SARS-CoV-2 Nsp1 regulates translation start site fidelity to promote infection" for consideration at PLOS Pathogens. As with all papers reviewed by the journal, your manuscript was reviewed by members of the editorial board and by several independent reviewers. In light of the reviews (below this email), we would like to invite the resubmission of a significantly-revised version that takes into account the reviewers' comments.

Specifically, the revised manuscript should address the following key questions raised by the reviewers:

(1) whether there is an effect on host mRNA levels upon infection and/or NSP1 expression to control for possible NSP1-mediated mRNA stability effects that have been previously reported.

(2) Address how specific these translational effects are to eIF1A versus other initiation components.

(3) whether the CUG start codon is indeed important for translation and, if needed, address why there is a discrepancy with previous reports about SL1 (which lacks the CUG) being sufficient to promote Nsp1-dependent translational increase.

(4) whether host RNAs that are selectively translated in the presence of NSP1 are enriched for uORFs in their 5’ UTRs.

In addition, I would encourage you to expand the Discussion to flesh out the significance of your findings as well as address some of the discrepancies between your observations and previous reports and what they might mean. This would also be a good opportunity to highlight what might be needed to resolve these questions in the future.

We cannot make any decision about publication until we have seen the revised manuscript and your response to the reviewers' comments. Your revised manuscript is also likely to be sent to reviewers for further evaluation.

Sincerely,

Mitchell Guttman

Guest Editor

PLOS Pathogens

Sara Cherry

Section Editor

PLOS Pathogens

Kasturi Haldar

Editor-in-Chief

PLOS Pathogens

orcid.org/0000-0001-5065-158X

Michael Malim

Editor-in-Chief

PLOS Pathogens

orcid.org/0000-0002-7699-2064

Reviewer's Responses to Questions

**Part I - Summary**

Reviewer #1: This manuscript applies an elegant combination of gradient fractionation-coupled mass spec, ribosome profiling, and IVT reporter-based translation experiments to address the question of how CoV2 influences the composition of polysomes and facilitates viral translation. While it is well established that Cov2 infection leads to host translational shutoff via the action of the viral nsp1 protein (which occludes 40S), it remains unclear how the virus may remodel the translation machinery to promote efficient CoV2 protein synthesis. In the first portion of the paper, Aviner et al. apply LC-MS/MS on sucrose gradient fractions of uninfected or CoV2 infected Vero and Calu3 cells, revealing several significant changes to the composition of primarily heavy polysome fractions upon infection. These include an increase in several proteostasis components, which when pharmacologically inhibited impair CoV2 replication, as well as translation initiation factors including EIF1A. In the second half of the paper, they turn their attention to the possible roles the altered polysome composition may play in translation of viral mRNA. They demonstrate that addition of the CoV2 genomic 5’ and 3’ leader sequences reduce reporter RNA translation efficiency, but that nsp1 is able increase viral translation, at least in part via recruitment of EIF1A to promote readthrough of an upstream, out of frame alternative start codon in the gRNA.

Overall, the data are of high quality and the connection between nsp1, EIF1A, and upstream CUG readthrough is an important advance.

Reviewer #2: In this manuscript, Aviner et al. use sucrose gradient fractionation and mass spectrometry to identify changes in the ribosome-associated proteome of cells infected with SARS-CoV-2. The authors identify increased association of specific translation factors, molecular chaperones and components of the 26S proteasome system with heavy polysome fractions in infected vs. mock-treated cells, indicating a drastic rewiring of the translation machinery upon SARS-CoV-2 infection. Importantly, the authors also show that a pharmacological inhibition of several of the identified factors reduces viral titers in a tissue culture model of SARS-CoV-2 infection, indicating that their proteomic study has identified new viral vulnerabilities that could serve as targets for future antiviral therapies.

In addition, the authors use reporter mRNAs to compare the translation of transcripts harboring cellular vs. viral UTRs. They find that reporter transcripts with viral UTRs are translated at lower rates, but that the overexpression of viral Nsp1 leads to a selective translation increase of reporter with viral, but not those with cellular UTRs. The authors ascribe the reduced translation of viral reporter mRNAs to the presence of an upstream non-cognate (CUG) start codon in the viral 5’ UTR, and show that a depletion of translation initiation factor eIF1A further increases ribosome footprints on the CUG codon, and abrogates the Nsp1-dependent translation boost of viral reporter mRNAs.

Overall, the study is well-written and thoroughly conducted. A few additional control are suggested below, to strengthen the proposed link between viral Nsp1, eIF1A and translation initiation from an upstream non-cognate start codon.

Reviewer #3: Aviner & Lidsky et al examined how protein synthesis and proteins associated with translating ribosomes (polysomes) were impacted upon infection by SARS-CoV-2. This is an important investigation because it remains unclear how the CoV2 gRNA and sgRNAs are translated upon the dramatic inhibition of host protein synthesis induced by the virus. The authors used subcellular proteomics, ribosome profiling, and reporter assays to identify that several molecular chaperones are preferentially associated with polysomes in CoV2 infected cells. With this information, they demonstrated that addition of several known inhibitors of the enriched chaperones inhibited replication of the virus. Furthermore, the authors identified that translation initiation factors (eIFs) are enriched in polysomes from infected cells, which is associated with decreased enrichment of 60S ribosomal proteins and elongation factors. This observation and complementary reporter assays in cells suggest that initiation on CoV2 gRNA is inefficient relative to a model host mRNA. The authors also demonstrate that the presence of NSP1 is needed to enhance translation of the CoV2 gRNA. By comparing their polysome data with published bio-ID data, the authors identify a potential link between NSP1 and eIF1A. Upon knockdown of eIF1A, translation of the viral RNA is reduced and ribosomes accumulate on an upstream CUG start site. eIF1A is also required for NSP1-mediated enhancement of viral RNA translation.

Overall, this study is interesting and well-written. It also provides important insight into how the translation of SARS-CoV-2 gRNA is mediated by NSP1 and host factors, in particular eIF1A. However, the manuscript should be improved by inclusion of additional experiments and changes.

**Part II – Major Issues: Key Experiments Required for Acceptance**

Reviewer #1: (No Response)

Reviewer #2: Major comments:

The authors present a model wherein a “functional interaction between Nsp1 and eIF4A1” reduces translation initiation on an upstream CUG start codon on the viral 5’ UTR, thereby selectively boosting the translation of viral transcripts. The model is intriguing, yet several controls are needed to strengthen this hypothesis.

Most importantly, previous studies have shown that an Nsp1-dependent translation boost is observed for reporter mRNAs carrying only SL1 of the viral UTR, yet this viral sequence does not contain the upstream CUG start codon. The authors should repeat the eIF1A depletion experiment with a reporter mRNAs carrying only the viral SL1 RNA to test if the functional interaction is indeed due to a repression of uORF translation. In addition, the authors should include a reporter mRNA in which the upstream CUG codon has been mutated.

According to the proposed model, Nsp1 overexpression should reduce translation initiation from the upstream CUG start codon. To strengthen this hypothesis, the authors should measure ribosome footprints on viral reporter RNA in control cells and cells overexpressing Nsp1 (ideally in the presence and absence of eIF1A).

If Nsp1 recruits eIF1A to repress uORF translation on viral transcripts, the same mechanism would likely affect cellular transcripts. Previous studies showed the selective translation of cellular RNAs in the presence of Nsp1. Are these transcripts enriched for uORFs in the 5’ UTR?

Additional comments:

- In Figure 5a the authors show that Nsp1 associates with polysomes, which is in contrast to previous studies that only found Nsp1 in 40S and 80S fractions of sucrose gradients. The authors should comment on this discrepancy. Is it likely due to higher sensitivity of the proteomic analysis, the formaldehyde crosslinking agents used in this study, or is it possible that the identified Nsp1 is part of nascent polypeptides synthesized on the ribosome?

In addition, the authors might reconsider the use of the word “significant” to describe the association of Nsp1 with polysome fractions, as no statistical test is shown to analyze the interaction.

- In Figure 5d the authors use qPCR to measure the amount of reporter mRNAs associated with polysomes in the presence or absence of Nsp1, yet they fail show total cellular reporter mRNA levels. This is a crucial control, as it was previously shown that Nsp1 can induce mRNA cleavage and degradation, and that mRNAs harboring viral 5’UTR sequences are partially resistant to Nsp1-induced RNA cleavage. In will be important to know if the preferred association of viral reporter mRNAs with polysomes in the presence of Nsp1 is due to selective recruitment, or because viral mRNA is stabilized while cellular mRNA is degraded.

- Figure 6d: The authors should include additional QC analysis of the ribosome profiling analysis, especially a graph showing the periodicity of the footprints in the ORF.

Reviewer #3: 1. In Figure 5c, the authors demonstrate that NSP1 enhances translation of the CoV2 gRNA. Given the destabilization of host mRNAs upon CoV2 infection or NSP1 expression, were there changes in the relative mRNA abundances in the presence of the viral proteins, which would confound interpretation of the translation data?

2. In Figure 6, knockdown of eIF1A increased initiation at an upstream CUG start site in CoV2 gRNA relative to the canonical start site for ORF1a. The authors should examine how knockdown of other components of the 43S PIC impact initiation on the viral RNA. For example, upon knockdown of eIF1, eIF2 and eIF3, given the cooperativity between eIFs during formation of the 43S PIC. Such experiments are important because they would clarify whether the observed effects are specific to eIF1A or represent a more generalizable outcome of having defective initiation complexes.

3. Does knockdown of eIF1A reduce the enrichment of other eIFs in the heavy polysome fraction upon CoV2 infection (i.e., how does eIF1A knockdown affect the enrichments reported in Fig. 4a)? Also, is the apparent lack of eIF2 enrichment in the heavy polysome fractions upon CoV2 infection (Figure 4a) a technical limitation or potentially relevant?

4. In Figure 6g, the authors show that the NSP1-mediated enhancement of gRNA translation by NSP1 required eIF1A. This is a very important figure in their manuscript, and it is missing a key component. The authors again should compare and/or report the mRNA levels of the reporter in each condition. Is it a translation effect or an RNA stability effect?

5. The authors cite one paper related to eIF1A function during initiation, which happens to support the authors’ observations. However, eIF1A has a multifaceted role and there has been substantial prior effort to understand its function both in vitro and in cells, in both mammals and yeast. The authors should place the findings they report here in the context of prior studies (e.g., foundational work by the Pestova, Hellen, Fraser, Hinnebusch, Dever, & Lorsch groups).

**Part III – Minor Issues: Editorial and Data Presentation Modifications**

Reviewer #1: 1. The paper appears to be written for a shorter format journal, and given that PPATH allows more text space, the significance of the findings would land with more impact if the authors fleshed out their central questions and experimental logic/interpretations more thoroughly. The figure legends also lack sufficient detail for easy understanding. For example, color coded labeling should be described in the legend.

2. Fig. 4f: please clarify how the composition (e.g., length, GC content) of the GAPDH UTR sequences compare to those of CoV2. In the spectrum of host mRNA translation efficiencies, where does GAPDH lie (is it average? highly efficient?).

3. The significance of the reduced level of nsp1 in polysomes relative to other ORF1a/b proteins is not clear (Fig. 5a). Why are Nsp2/3/4/8/10 polysome associated? If they are there simply because they are in the process of being translated (rather than for a regulatory reason), is nsp1 relatively absent from polysomes because it is released more rapidly from the polyprotein than the others?

4. A key finding is that depletion of EIF1A prevents nsp1 from boosting the translation efficiency of the CoV2 leader-containing RNA (Fig6f-g). I assume this mechanism should only work on viral genomic RNA, not subgenomic RNA (indicated in Fig.6F). Yet, other reports have shown that reporters containing the CoV2L derived from sgRNA escape nsp1 repression in transfection experiments. More discussion is warranted on this point, e.g., is it a difference between escaping repression and translational enhancement? Additionally, it would be valuable to include a version of the luc reporter where the upstream CUG is mutated in the Fig.6g assay, to confirm the extent to which the role of EIF1A involves readthrough of that codon.

Reviewer #2: - Extended Figure 1b – legend is missing. Please describe what is shown with the orange and grey line. Please use different color coding, since grey is used for mock-infected cells throughout the manuscript.

- Figure 3: label for panel “d” is missing.

- Figure 6g: The authors should show absolute luciferase activity, not just normalized data, to demonstrate how much the depletion of eIF1A changes the overall translation of all reporter constructs.

Reviewer #3: 1. Extended data Fig 1e is a bit unexpected. Why are there more RPs present in light polysomes relative to mock, despite identical polysome curves? The data do not appear to support this statement: “The abundance of ribosomal proteins in each pooled fraction, as quantified by MS (Fig. 1f-g and Extended Data Fig. 1d), was consistent with the observed rRNA absorbance profiles (Fig. 1d).”

2. In the sucrose gradients used for the polysome fractionation experiments, where does the CoV2 gRNA migrate? It was hinted at early in the text, but does it migrate into the heavy polysome fractions? It could be helpful to analyze migration of in vitro transcribed gRNA on the gradient to determine this. This is relevant mechanistically because it might help distinguish whether a viral gRNA contains a single, paused 40S initiation complex, or whether the gRNA contains 80S ribosomes and an initiation 40S complex.

3. In figure 4f, does removing the uORF yield similar translation as GAPDH? In other words, is it the uORF or the complex secondary structure that limits translation of the gRNA? How does gRNA translation compare to sgRNA translation?

4. As far as we are aware, all structures of the NSP1-bound ribosome do not contain mRNA. So, this statement is not supported by the published data: “This suggested that Nsp1 may preferentially interact with or stabilize 48S pre-initiation complexes with specific composition or function, as supported by cryo-EM studies of Nsp1-bound ribosomes.”

5. “It is tempting to speculate that Nsp1 stabilizes eIF1A binding or otherwise affects its interactions with the ribosome decoding center, although additional work is needed to elucidate the specifics of the mechanism.” This is unlikely given this preprint (Universal features of Nsp1-mediated translational shutdown by coronaviruses), which shows that SARS-CoV-2 NSP1 does not impact eIF1A association or dissociation from 40S subunits, albeit in the absence of other eIFs.

6. In figure 6g, what happens if you remove the CUG(60) on CoV2? Does NSP1 lose all stimulatory effects on translation relative to GFP? How is translation then impacted after transfecting with Si-1A?

7. From figure 6a and figure 4a, it appears RNA helicases eIF4A1 and DDX3 both associate to NSP1 and are recruited to heavy polysome fractions during CoV2 infection. Complex RNA secondary structure has been shown to promote upstream CUG translation initiation, suggesting helicases could be important for translation of downstream ORFs. It would be interesting to see if knockdown of those two helicases also decreases NSP1-mediated stimulation of translation on CoV2 gRNA.

PLOS authors have the option to publish the peer review history of their article (what does this mean?). If published, this will include your full peer review and any attached files.

Reviewer #1: No

Reviewer #2: No

Reviewer #3: No
---

## [Decision Letter · Decision Letter 1]

15 Dec 2023

Dear Dr. Andino,

Thank you very much for submitting your manuscript "SARS-CoV-2 Nsp1 regulates translation start site fidelity to promote infection" for consideration at PLOS Pathogens. As with all papers reviewed by the journal, your manuscript was reviewed by members of the editorial board and by several independent reviewers. The reviewers appreciated the attention to an important topic. Based on the reviews, we are likely to accept this manuscript for publication, providing that you modify the manuscript according to the review recommendations.

There is one significant technical concern raised. Specifically, Reviewer #2 is concerned about whether there is protection from mRNA degradation of the reporter in the presence of NSP1 and whether the transfection methods used might impact the conclusions. We would appreciate if you could address this either by providing additional data or by explaining why this is not a concern.

In addition, Reviewer #2 makes an important suggestion about rewording the claim about the role of NSP1 in start codon selection to state that eIF1A appears to regulate the Nsp1-dependent translation boost. We would encourage you to include this or similar language in the manuscript to note this remaining uncertainty.

Sincerely,

Mitchell Guttman

Guest Editor

PLOS Pathogens

Sara Cherry

Section Editor

PLOS Pathogens

Kasturi Haldar

Editor-in-Chief

PLOS Pathogens

orcid.org/0000-0001-5065-158X

Michael Malim

Editor-in-Chief

PLOS Pathogens

orcid.org/0000-0002-7699-2064

The revised manuscript has now been reviewed by 2 of the initial reviewers and they both believe the manuscript is significantly improved.

Before we can accept the manuscript for publication, there is one significant technical concern raised that we would like you to address. Specifically, Reviewer #2 is concerned about whether there is protection from mRNA degradation of the reporter in the presence of NSP1 and whether the transfection methods used might impact the conclusions. I would appreciate if you could address this concern either by providing additional data or by explaining why this is not a concern.

In addition, Reviewer #2 makes an important suggestion about rewording the claim about the role of NSP1 in start codon selection to state that eIF1A appears to regulate the Nsp1-dependent translation boost. I would encourage you to include this or similar language in the manuscript to note this remaining uncertainty.

Reviewer Comments (if any, and for reference):

Reviewer's Responses to Questions

**Part I - Summary**

Reviewer #1: The authors have nicely addressed all of my prior concerns.

Reviewer #2: The authors have substantially revised and significantly improved the manuscript. In particular, the analysis of ribosome footprinting data of Nsp1-deficient and WT viruses strongly strengths the presented model.

However, the revision failed to sufficiently address 2 of my previous concerns (outlined below, related to the role of Nsp1 in regulating RNA stability and the direct connection between Nsp1 and uORF translation). In the current form, the presented data is insufficient to support the model that Nsp1 regulates eIF1A-dependent start codon selection on viral RNAs. An additional round of minor revisions is therefore suggested, before the manuscript is ready for publication.

**Part II – Major Issues: Key Experiments Required for Acceptance**

Reviewer #1: (No Response)

Reviewer #2: 1. One of the major concerns was that total reporter mRNA levels were not measured in the previous version of this manuscript. It was therefore impossible to determine whether the viral 5'UTR protected from Nsp1-dependent RNA degradation or Nsp1-dependent translation repression. In the revised manuscript, the authors include total RNA quantification, and conclude that luciferase reporter mRNA levels are unchanged. However, the authors use lipofectamine for direct mRNA transfection, which makes a quantification of *accessible* cytoplasmic mRNA impossible. It is very likely that a majority of transfected mRNA is still contained in lipid vesicles within the cell, and therefore protected from degradation. A different transfection method (e.g. electroporation) or DNA transfection is needed to draw conclusions about RNA stability. Without additional experiments, the authors would need to weaken their statement to convey that, based on the presented data, they cannot exclude a primary effect of the 5'UTR on RNA stability.

2. While the authors present a vast amount of data supporting the hypothesis that viral uORF translation is regulated by eIF1A (and eIF1), the link between Nsp1 and uORF translation is still weak. The only direct evidence comes from Fig. 6g, which shows that eIF1A-depletion reduces the Nsp1-dependent boost in viral reporter mRNA translation. While this experiment shows a link between eIF1A and Nsp1, it does not show that this link is dependent on the upstream CUG start codon. The CUGAUG mutant is insufficient, because it almost completely abolishes all translation. If the authors want to make the statement that Nsp1 regulates start codon usage via eIF1A, they should repeat the experiment of Fig. 6g with a reporter that does not have a upstream start codon at all. This is particularly relevant as the Nsp1-dependent translation boost has been shown for reporter mRNAs that only carry the viral SL1, and no upstream start codon. Alternatively or additionally, changes in ribosomal foot prints on upstream start codons in the presence and absence of Nsp1 (similar to Figure 6c) could be shown to strengthen the link between Nsp1 and start codon selection. In the absence of either of these important experiments, the authors should should not conclude that Nsp1 regulates start codon selection, they can merely state that eIF1A appears to regulate the Nsp1-dependent translation boost.

**Part III – Minor Issues: Editorial and Data Presentation Modifications**

Reviewer #1: (No Response)

Reviewer #2: The statement "A conserved stem-loop (SL1) found at the 5’ end of all viral RNAs (Fig. 6a) displaces Nsp1 from the ribosome" should be revised. There is currently no direct evidence that SL1 directly displaces Nsp1 from the ribosome, only that SL1 containing RNAs escape the Nsp1-dependent repression.

PLOS authors have the option to publish the peer review history of their article (what does this mean?). If published, this will include your full peer review and any attached files.

Reviewer #1: No

Reviewer #2: No

Figure Files:

Data Requirements:

Reproducibility:

References:

---

## [Editor Report · Decision Letter 2]

8 Jan 2024

Dear Dr. Andino,

We are pleased to inform you that your manuscript 'SARS-CoV-2 Nsp1 cooperates with initiation factors EIF1 and 1A to selectively enhance translation of viral RNA' has been provisionally accepted for publication in PLOS Pathogens.

Best regards,

Mitchell Guttman

Guest Editor

PLOS Pathogens

Sara Cherry

Section Editor

PLOS Pathogens

Kasturi Haldar

Editor-in-Chief

PLOS Pathogens

orcid.org/0000-0001-5065-158X

Michael Malim

Editor-in-Chief

PLOS Pathogens

orcid.org/0000-0002-7699-2064
---

## [Editor Report · Acceptance letter]

5 Feb 2024

Dear Dr. Andino,

We are delighted to inform you that your manuscript, "SARS-CoV-2 Nsp1 cooperates with initiation factors EIF1 and 1A to selectively enhance translation of viral RNA," has been formally accepted for publication in PLOS Pathogens.

Best regards,

Michael Malim

Editor-in-Chief

PLOS Pathogens

orcid.org/0000-0002-7699-2064